# Profiling subcellular localization of nuclear-encoded mitochondrial gene products in zebrafish

Barbara Uszczynska-Ratajczak[1,2,*] , Sreedevi Sugunan[3,4,*], Monika Kwiatkowska[1,2,4] , Maciej Migdal[4],
Silvia Carbonell-Sala[5], Anna Sokol[6,7], Cecilia L Winata[4] , Agnieszka Chacinska[8]

**Most mitochondrial proteins are encoded by nuclear genes, synthetized in the cytosol and targeted into the organelle. To characterize the spatial organization of mitochondrial gene products in zebrafish (*Danio rerio*), we sequenced RNA from different cellular fractions. Our results confirmed the presence of nuclear-encoded mRNAs in the mitochondrial fraction, which in unperturbed conditions, are mainly transcripts encoding large proteins with specific properties, like transmembrane domains. To further explore the principles of mitochondrial protein compartmentalization in zebrafish, we quantified the transcriptomic changes for each subcellular fraction triggered by the *chchd4a*$^{-/-}$ mutation, causing the disorders in the mitochondrial protein import. Our results indicate that the proteostatic stress further restricts the population of transcripts on the mitochondrial surface, allowing only the largest and the most evolutionary conserved proteins to be synthetized there. We also show that many nuclear-encoded mitochondrial transcripts translated by the cytosolic ribosomes stay resistant to the global translation shutdown. Thus, vertebrates, in contrast to yeast, are not likely to use localized translation to facilitate synthesis of mitochondrial proteins under proteostatic stress conditions.**

## Introduction

Despite the presence of mitochondrial DNA, the vast majority of mitochondrial proteins (~99%) are encoded by the nuclear genome. Therefore, the biogenesis of functional mitochondria relies on the synthesis and import of those proteins, in their precursor form, into the organelle. Efficient protein import through a highly complex mitochondrial structure requires a system regulating transport and assembly of precursor proteins (Neupert & Herrmann, 2007;

Chacinska et al, 2009; Fox, 2012). After the synthesis on cytosolic ribosomes, precursor proteins are imported through the translocase of the membrane (TOM) complex, which is a common entry gate for all mitochondria-destined precursor proteins. After crossing TOM, precursor proteins are directed to specific mitochondrial compartments including outer mitochondrial membrane (OMM), inner mitochondrial membrane (IMM), intermembrane space (IMS), and matrix (Bolender et al, 2008; Endo et al, 2011; Becker et al, 2012).

Although the mitochondrial protein import has been extensively studied for decades, the cytosolic stage of this process is not fully understood (Avendaño-Monsalve et al, 2020). At the same time, an efficient trafficking of RNA into specific compartments is a fundamental task of eukaryotic cells, as a precise distribution of mRNAs across organelles in the cell allows to synthetize proteins exactly, where they are needed (Buxbaum et al, 2015). As a consequence, numerous studies have indicated the presence of mRNAs encoding mitochondrial proteins, either on the mitochondrial surface or in its close proximity (Suissa & Schatz, 1982; Egea et al, 1997; Corral-Debrinski et al, 2000; Diehn et al, 2000; Gadir et al, 2011). Moreover, mitochondria-associated mRNAs have been proven to be active templates for protein synthesis (Marc et al, 2002), whereas the cytosolic ribosomes observed at OMM (Crowley & Payne, 1998; Ni et al, 1999) are able to interact with the TOM complex (Gold et al, 2017). Thus, in living cells, the translation of nuclear-encoded proteins can occur at the surface of mitochondria and might be directly coupled to their import into this organelle (Knox et al, 1998; Marc et al, 2002). Interestingly, it seems that the physicochemical properties of mitochondrial proteins, including hydrophobicity or folding speed, might determine the protein import route (Sass et al, 2003; Liu & Liu, 2007; Yogev et al, 2007; Williams et al, 2014). Despite these findings, mitochondrial protein import remains almost completely unexplored for higher eukaryotes, as this process was mainly studied using yeast (Suissa & Schatz, 1982; Egea et al, 1997; Corral-Debrinski et al, 2000; Diehn et al, 2000; Marc et al, 2002; Gadir et al, 2011; Lesnik et al, 2014,

[1]Institute of Bioorganic Chemistry, Polish Academy of Sciences, Poznan, Poland    [2]Centre of New Technologies, University of Warsaw, Warsaw, Poland    [3]ReMedy International Research Agenda Unit, University of Warsaw, Warsaw, Poland    [4]International Institute of Molecular and Cell Biology, Warsaw, Poland    [5]Centre for Genomic Regulation, The Barcelona Institute of Science and Technology, Barcelona, Spain    [6]Department of Developmental Genetics, Max Planck Institute for Heart and Lung Research, Bad Nauheim, Germany    [7]Biomolecular Mass Spectrometry, Max Planck Institute for Heart and Lung Research, Bad Nauheim, Germany    [8]ReMedy International Research Agenda Unit, IMol Polish Academy of Sciences, Warsaw, Poland

Correspondence: a.chacinska@imol.institute; cwinata@iimcb.gov.pl; buszczynska@ibch.poznan.pl
*Barbara Uszczynska-Ratajczak and Sreedevi Sugunan contributed equally to this work.

2015; Williams et al, 2014; Tsuboi et al, 2020) and mammalian cell (Matsumoto et al, 2012; Gehrke et al, 2015; Fazal et al, 2019) models.

Biogenesis of mitochondria is a precisely controlled process that largely depends on a fine-tuned balance between synthesis and degradation of many cellular proteins (Topf et al, 2019). Disorders in the mitochondrial protein import can trigger serious consequences for the cell, including energetic deficiencies and proteostatic stress response induced by the accumulation of precursor proteins in the cytosol (Wang & Chen, 2015). Although there are various cellular mechanisms that can reduce the accumulation of mis-targeted mitochondrial proteins (Wang & Chen, 2015; Wrobel et al, 2015; Topf et al, 2018), we still do not understand the impact of mitochondrial stress on the subcellular localization of mRNAs encoding mitochondrial proteins. In general, the knowledge on how a cell maintains the biogenesis of mitochondria under the proteostatic stress is very limited. At the same time, various alternative mitochondrial protein targeting routes that protect and redirect mistargeted proteins outside mitochondria exist (Itakura et al, 2016). Surprisingly, many of them involve interaction with ER (Hansen et al, 2018; Xiao et al, 2020 Preprint), in particular under proteostatic stress conditions (Gamerdinger et al, 2015, 2019; Matsumoto et al, 2019).

Here, we studied the subcellular localization of nuclear-encoded mitochondrial mRNAs in zebrafish to uncover transcripts that are enriched and thus, are likely to be translated on the mitochondrial surface. In addition, this study for the first time reports the dynamics of subcellular localization of mitochondrial transcripts in physiological conditions and under protein import deficiency. To assess the impact of mitochondrial protein import disorders on the mitochondria-associated transcripts, we took advantage of $chchd4a^{-/-}$ zebrafish mutant line (Sokol et al, 2018). As previously described, loss of function of the $chchd4a$ paralogue affects the activity of the MIA pathway—an essential route for import and assembly of mitochondrial proteins in the IMS. Here, we further explore the transcriptomic response triggered by $chchd4a^{-/-}$ mutation within the context of subcellular mRNA localization. Taken together, this study broadens our knowledge about the principles and dynamics of distributing mRNAs encoding mitochondrial proteins across the cell in vertebrates.

# Results

### Development and validation of subcellular fractionation method

To investigate which nuclear-encoded mitochondrial transcripts are likely to be located on the surface of mitochondria and hence participate in localized translation (Fig 1A), we established a new protocol for biochemical fractionation (Fig 1B) in zebrafish larvae. Although this approach largely adopts a strategy that was used to obtain intact mitochondria with associated ribosomes in yeast (Gold et al, 2017), it was optimized to obtain a membrane-bound fraction containing mitochondria and ER together with their associated ribosomes. Preservation of ER in this fraction allows us to perform more global analysis of membrane associated nuclear-encoded mitochondrial transcripts, as ER and mitochondria are highly interconnected in the cell. This not only includes their

physical connections through mitochondria-associated membranes (Missiroli et al, 2018), but also their extensive interactions supporting mitochondrial protein import (Gamerdinger et al, 2015, 2019; Hansen et al, 2018; Matsumoto et al, 2019; Xiao et al, 2020 Preprint).

Standard biochemical strategies for isolating mitochondria use EDTA as one of the protease inhibitors in the isolation buffer (IB-E). EDTA disrupts association of cytosolic ribosomes with mitochondria by chelating the metal ions that keep the ribosomes intact. Therefore, for the isolation of mitochondria with associated ribosomes, a ribosome-friendly buffer (IB-M) containing $MgCl_2$, cycloheximide (CHX), and RNAse inhibitors was used. Magnesium ions are essential for the integrity of ribosomes (Shenvi et al, 2005), whereas CHX stabilizes ribosome-nascent chain complexes. By homogenization and differential centrifugation using IB-E or IB-M, we obtained three subcellular fractions – membrane-bound (MB), high-speed (HS) and whole, unfractionated cell (total) (Fig 1B). Analysis of subcellular fractionation by Western blotting showed that the membrane-bound fraction was exclusively enriched with intact mitochondria as indicated by the mitochondrial markers, including proteins from various subcompartments—Tomm20 of OMM, Cox4i1 of IMM, and Timm9 of IMS (Fig S1E, lane 1–3). These markers were not visible in the high-speed and whole-cell fractions. The ER marker calreticulin (Calr) was also enriched in the membrane-bound fraction compared with unfractionated cells and high-speed fraction obtained using both buffer systems (Fig S1E, lane 4). Interestingly, parts of ER were also detected in the HS fraction, which means that because of its size and diverse structure, ER splits between two fractions. However, the HS fraction showed no enrichment of mitochondrial markers. In addition, both high-speed and membrane-bound fractions obtained with IB-M were enriched in ribosomal proteins (Fig S1E, lane 5 and 6). In contrast, the cytosolic fraction obtained with IB-M did not show any ribosomal protein enrichment. We reasoned that this could be due to the co-sedimentation of free polysomes with HS fraction at very high speed used during ultracentrifugation. Consequently, the HS fraction contains a combination of both ER-bound ribosomes and free cytosolic polysomes. We also noticed that Gapdh, the cytosolic protein, was mainly enriched in the cytosolic fraction (Fig S1E, lane 7).

After determining the composition of subcellular fractions, we isolated MB and HS fractions with IB-M containing BSA. BSA helps to preserve function and integrity of mitochondria and is expected to enable reliable profiling of MB-associated RNAs (Meisinger et al, 2000; Gold et al, 2017). It actually neutralizes the negative action of fatty acids activated by phospholipases during cell disruption, which inhibits proper function of mitochondria by uncoupling oxidative phosphorylation (OXPHOS) (Di Paola & Lorusso, 2006; Ngo et al, 2021). At high concentration, BSA may protect mitochondrial proteins from degradation, serving as an alternative substrate for proteases released upon cell breakage (Goldstein et al, 1981). The comparison between the IB-M isolated and the EDTA-stripped MB fractions (isolated with IB-E containing BSA), once again proved the presence of intact mitochondria and enrichment of cytosolic ribosomal subunits in the IB-M isolated fraction (Fig 1C). Analysis of the HS fraction also confirmed the enrichment of cytosolic ribosomes in comparison with its EDTA-stripped counterpart. In addition, an abundant presence of ER in the IB-M isolated MB was noticed in

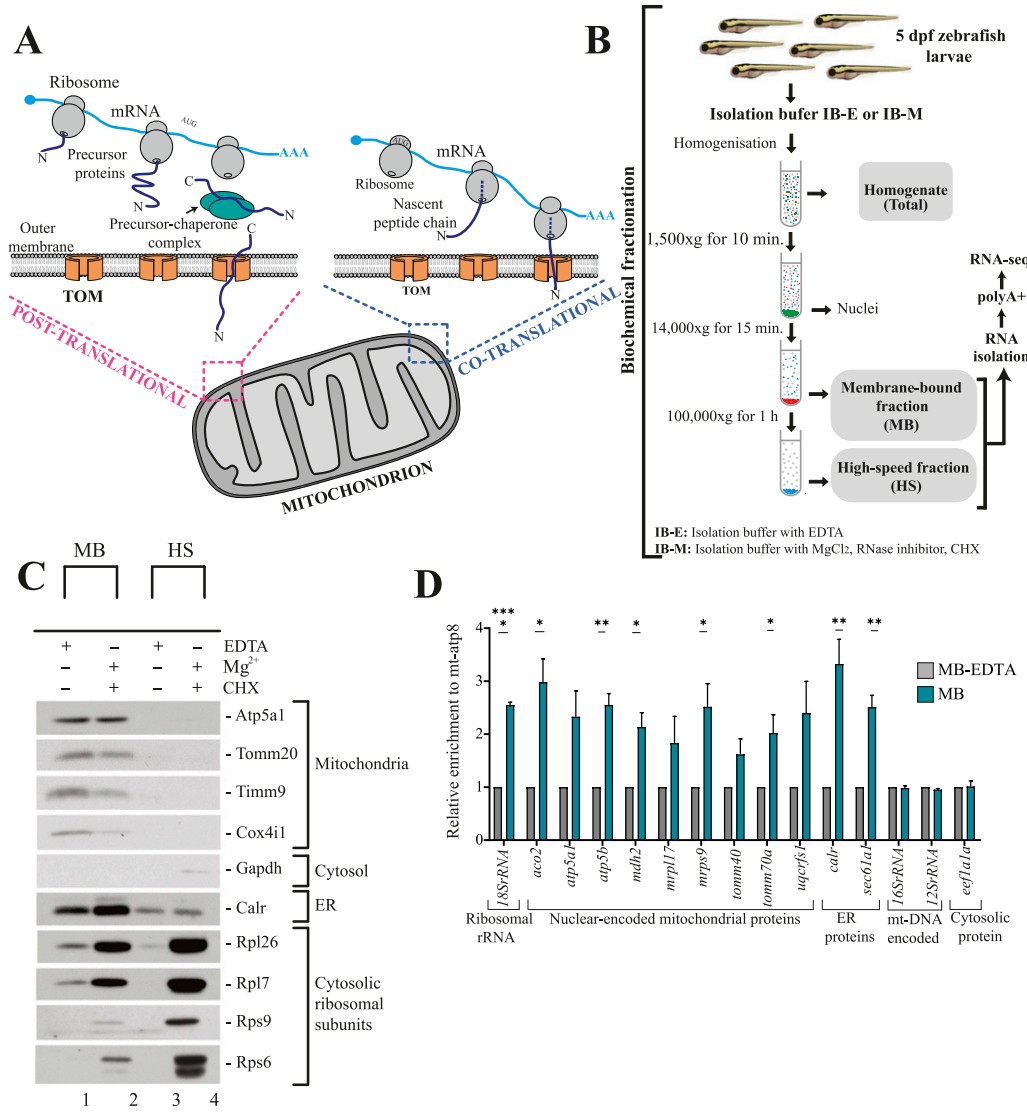

**Figure 1. Isolation and identification of membrane associated mRNAs.**
**(A)** Routes for mitochondrial protein import. In post-translational import, proteins are imported to mitochondria after their complete synthesis in the cytosol in a process aided by chaperones and co-chaperones. Whereas the localized translation that occurs at the surface of mitochondria may be directly coupled with the translocation of proteins into organelle. **(B)** Biochemical fractionation of 5 days post fertilization zebrafish larvae based on strategy to obtain intact mitochondria with associated ribosomes in yeast (Gold et al, 2017). Isolation of mitochondria with associated ribosomes was performed using a ribosome-friendly buffer (IB-M) containing MgCl2, cycloheximide, and RNAse inhibitors. MB and HS fractions were obtained via differential centrifugation. Total RNA isolated from each fraction was enriched for polyadenylated fraction and subjected to short-read sequencing using Illumina platform. **(C)** Isolation of membrane-bound and high speed fractions. **(D)** qRT-PCR validation of MB fraction. Relative enrichment of genes in MB with respect to EDTA-stripped MB upon normalization with mitochondrial DNA encoded mt-atp8 is shown. Data derived from three biological replicates. Error bars correspond to SEM; $P < 0.05$ (*), $P < 0.01$ (**), and $P < 0.0001$ (****) by unpaired $t$ test.

comparison with the EDTA-stripped MB fraction, whereas the IB-M isolated and EDTA-stripped HS fractions showed similar ER enrichment. The IB-M isolated MB and HS fractions were used for all subsequent experiments, and henceforth we will refer to them as MB and HS.

To prove that the MB fraction is enriched with nuclear-encoded mitochondrial transcripts, relative RNA levels of the MB with respect to the EDTA-stripped MB were determined by the qRT-PCR analysis (Fig 1D). This analysis included mainly zebrafish orthologues of nuclear-encoded mitochondrial transcripts known to be localized

on the surface of mitochondria in yeast (Williams et al, 2014). Five of nine of these nuclear-encoded mitochondrial mRNAs were significantly enriched in the MB with respect to its EDTA-stripped counterpart. Whereas, the levels of mt-DNA encoded transcripts and the *eef1a1a* transcript encoding a cytosolic protein remained similar between MB and EDTA-stripped MB fraction. In line with the previous observations pointing towards the enrichment of ER fragments in the MB, the *sec61a1* and *calr* transcripts encoding ER membrane and the ER luminal proteins, respectively, were also found to be enriched in the MB fraction. We also confirmed the

enrichment of 18S rRNA—the structural component of the cytosolic ribosome in this fraction.

### High-throughput profiling of subcellular localization of mitochondrial gene products

To examine the subcellular localization of nuclear-encoded mitochondrial transcripts, we isolated polyadenylated RNA from MB and HS fractions obtained via biochemical fractionation from WT zebrafish individuals. We used short-read Illumina RNA sequencing to identify genes with transcripts enriched in each fraction. To rank transcripts from the most membrane associated to those most likely translated by the free cytosolic polysomes, we directly compared two fractions with each other. The HS fraction was a reference in these comparisons, so all genes that are enriched in this analysis are likely to be membrane bound (MB-enriched) and those that are depleted are more likely to be translated by free cytosolic polysomes (HS-enriched). Interestingly, we observed two populations of genes (Fig S2A) enriched either in HS or MB. We did not detect genes that would be present to the same extent in both fractions. One of the possible reasons for this, could be exclusion of poorly enriched genes from the analysis (see the Materials and Methods section for details). In total, the differential analysis showed enrichment of 2,177 and 3,315 genes in MB and HS fractions, respectively (Fig 2A). To reliably assign genes to each fraction, we required at least twofold enrichment (highly enriched) between MB and HS fractions (FDR 5%; $\log_2 FC \Leftrightarrow \pm 1$). As this threshold was selected arbitrarily to match the common standards of RNA-seq data analysis, we also distinguished a class of moderately enriched genes (FDR 5%; $0.5 \leq \log_2 FC < 1$ or $-1 < \log_2 FC \leq -0.5$). However, this group was much less numerous with 1,455 and 1,316 genes enriched in MB and HS fractions (Fig 2A). We also observed a substantial enrichment of all 13 genes encoded by the mitochondrial genome among the MB highly enriched genes (Fig S2B), which confirms the presence of mitochondria in this fraction.

To define a leading group of proteins that are likely to be synthesized on the surface of mitochondria, we performed Gene Ontology (GO) enrichment analysis. This analysis showed highly MB-enriched genes to be involved in transmembrane transport, in particular ion and anion transport (Fig 2B). To further expand this analysis, we switched the GO aspect to the "cellular component." The cellular component search revealed many of MB-enriched genes to be integral components of membranes, especially plasma membranes (Fig S2C). Whereas, genes enriched in the HS fraction were mainly linked to translation and peptide metabolic processes (Fig 2B), which were additionally described as small and large ribosomal units by the cellular component aspect (Fig S2C). To further expand the global examination of gene groups detected in each fraction, we performed an enrichment analysis using the Kyoto Encyclopaedia of Genes and Genomes (KEGG). This analysis revealed ATP-binding cassette (ABC) transporters as one of the over-represented KEGG terms among highly MB-enriched genes (Fig S2D). Interestingly, highly HS-enriched genes were reported to be either involved in the OXPHOS or encoding ribosomal proteins. Similar GO terms, as for highly enriched genes, were also identified for moderately enriched ones, including both biological process (Fig S2E) and cellular component (Fig S2F) GO aspects for MB and HS

fractions, respectively, whereas no enriched KEGG pathways were detected for moderately enriched genes in MB and HS fractions.

Next, we carefully investigated the subcellular localization of mitochondrial gene products by using merged set of zebrafish orthologues for two independent human gene inventories with strong support of mitochondrial localization: MitoCarta 2.0 (Calvo et al, 2016) and Integrated Mitochondrial Protein Index (http://www.mrc-mbu.cam.ac.uk/impi). This analysis revealed 80 (73 + 7) and 249 (243 + 6) mitochondrial genes to be highly enriched in the MB and HS fractions, respectively (Figs 2C and S3A), indicating that nuclear-encoded mitochondrial genes are mainly present in the HS fraction. We noticed the same trend for moderately enriched genes with 91 and 182 mitochondrial genes detected in MB and HS fractions, respectively (Figs 2C and S3B). To investigate the enrichment of ER genes in each fraction, we used zebrafish orthologues of 483 (~2% of all protein coding human genes) human genes that encode proteins localizing to the ER (Table S1), as shown by the Human Protein Atlas (Uhlén et al, 2015). Our results indicate that the ER genes are uniformly enriched across all fractions, with an exception of only 13 detected in the moderately enriched gene set for the HS fraction. Genes showing dual localization (Fig 2C) were included in both ER and mitochondrial gene sets, respectively.

To further explore the presence of mitochondrial genes in HS and MB fractions, we analyzed the distributions of enrichment values for different mitochondrial compartments (Fig 2D). Most genes for each mitochondrial compartment have their transcripts enriched in the HS, whereas individual cases such as *shmt2* gene (matrix) involved in the one-carbon pathway (Ducker & Rabinowitz, 2017) are enriched in the MB fraction. The transcripts encoding IMS proteins are distinct from others subgroups, as they were almost entirely localized in the HS fraction. Importantly, the MB enrichment observed for the IM proteins is solely driven by the enrichment of mitochondrially encoded genes (Fig S2B). According to our previous observations, ER genes appeared to be uniformly enriched across investigated fractions, which is in line with estimated fraction composition (Fig 1C).

We further compared observed gene enrichments with the outcome of other studies investigating the subcellular localization of mitochondrial gene products, mainly in yeast (Diehn et al, 2000; Marc et al, 2002; Fox, 2012) and human cell lines (Fazal et al, 2019). Using our data, we analyzed the enrichment of genes with transcripts known to be located in the proximity of mitochondria, translated by mitochondria-bound or free cytosolic polysomes. Before this analysis, we identified a zebrafish orthologue for each of those genes. Interestingly, only two yeast genes *ALD4* and *MCR1* for which transcripts are known to be translated by the mitochondria-bound polysomes (Diehn et al, 2000; Marc et al, 2002) were highly enriched, whereas the other two *ATM1* (Corral-Debrinski et al, 2000) and *PDA1* (Marc et al, 2002) were moderately enriched in the MB fraction (Fig 2E). On the other hand we observed a much higher validation rate for genes known to be translated by the free cytosolic polysomes, as 11 and 6 of 19 investigated genes showed high and moderate enrichment in HS fraction, respectively. To clarify the poor overlap between our membrane-associated transcripts and those translated by the mitochondria-bound polysomes in yeast, we explored the enrichment of zebrafish orthologues of human genes, which had their products detected in the close proximity of

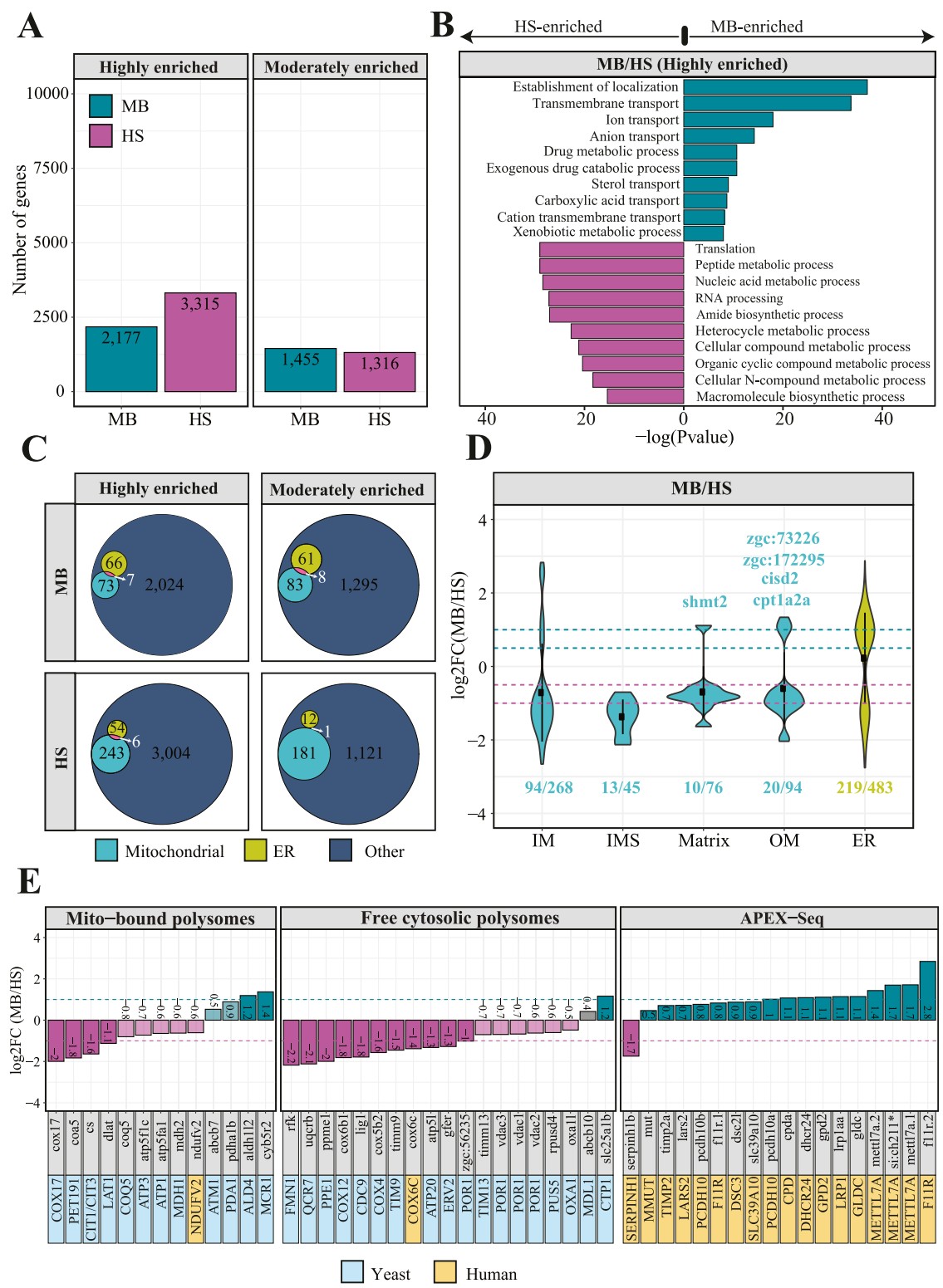

**Figure 2. Subcellular localization of nuclear-encoded mitochondrial mRNAs.**
**(A)** Bar plots representing the total number of genes identified in MB and HS fractions obtained via biochemical fractionation of WT 5 dpf zebrafish samples and short-read Illumina RNA sequencing. **(B)** Gene Ontology (GO) enrichment for highly enriched genes in MB (2,177) and HS (3,315) fractions using biological process aspect. The results are shown as a negative log$_{10}$ P-value after Bonferroni correction. Bars in green indicate GO terms enriched for genes detected in MB fraction, whereas the purple bars represent the GO terms for the features identified in HS. For better representation of the results the bars enriched for genes in HS were transformed to be located on the left hand side of the plot. **(C)** Venn diagrams representing the number of mitochondrial genes from the merged MitoCarta 2.0 (Calvo et al, 2016) and IMPI repository, as well as the zebrafish orthologues of human 483 genes encoding proteins that localize to ER according to the Human Protein Atlas, identified in MB or HS fraction from WT

OM by APEX-Seq study (Fazal et al, 2019). In this comparison, 94% (17/18) genes were also enriched in our MB fraction (Fig 2E, panel 3). This could suggest that the discrepancy between the enrichment of transcripts located at the surface of mitochondria in zebrafish and yeast is driven by the differences in the organismal complexity.

To explore this finding in detail, we compared our enrichment scores to the mitochondrial enrichment in yeast obtained using the high-throughput proximity specific ribosome profiling (Williams et al, 2014). We observed poor correlation between the MB-enrichment in zebrafish and the mitochondrial enrichment reported by the Weiss-man group in both absence (Fig S4A) and presence (Fig S4B) of CHX – a translation elongation inhibitor. Interestingly, the overlap between genes localizing to the mitochondrial surface in yeast and zebrafish is higher for the OM45 enrichment reported in absence of CHX. In addition to many mitochondrially encoded genes, we noticed the enrichment of genes encoding ABC transporters, such as *MDL1* and *MDL2* in yeast, both of which correspond to *abcb11a*, *abcb11b*, and *tap1* (*transporter 1*, *ABC*, *sub-family B*) genes in zebrafish (Fig S4A). These are also the most enriched gene pairs detected in the presence of CHX (Fig S4B). In general, we observed more HS-enriched zebrafish genes in the comparison with CHX treated yeast samples. Moreover, five of them appeared as top enriched in both CHX treated and untreated cells. These include *PUF3* (*PUmilio-homology domain Family*) gene encoding protein of the mitochondrial outer surface in yeast and *pum2* (*pumilio RNA-binding family member 2*) gene—its zebrafish orthologue, encoding protein also predicted to have mRNA 3'-UTR binding activity. Interestingly, *pum2* is predicted to localize to cytosol (Wang et al, 2012b). Finally, a thorough analysis of genes that are mitochondrially enriched in yeast (in absence of CHX) and MB-enriched in zebrafish revealed the presence of cases involved in ion transport (Fig S4C). This includes *mitochondrial metal transporter 1* (*MMT1*) and *mitochondrial metal transporter 2* (*MMT2*)—two paralogues involved in mitochondrial iron movement in yeast (Lange et al, 1999), corresponding to two solute careers *slc30a1a* and *slc30a8* in zebrafish. The major difference between genes enriched in CHX-treated and CHX-untreated yeast cells are the changes observed for the mitochondrially encoded genes, which were globally depleted in the presence of CHX (Fig S4D). This could be the effect of coordinated activity of the nuclear and mitochondrial genomes, which by putting expression of mitochondrially encoded genes into the cellular context can result in silencing of some of them (Cruz-Zaragoza et al, 2021).

### Properties of membrane-associated mitochondrial transcripts

To better understand the principles determining the specific cellular localization of transcripts encoding mitochondrial proteins, we investigated the enrichment of specific gene families. We started this analysis by looking at transmembrane transport, as suggested by the GO (Figs 2B and S2D) and KEGG pathway (Fig S2D) enrichment analysis. Mitochondria to perform their biological functions require importing and exporting various solutes and metabolites. Although

the OMM is quite permeable, the real challenge is to cross the IMM. To overcome the problem of intrinsically impermeable IMM, mitochondria use highly specific transporters that can be divided into four major families: the solute carriers (SLCs), ABC transporters, the mitochondrial pyruvate carrier (MPC), and sideroflexin (SFXN) carriers (Cunningham & Rutter, 2020).

The ABC transporters are ubiquitous membrane-bound proteins that are also ATP-dependent pumps, allowing the movement of substrates in or out of cells (Vasiliou et al, 2009). All ABC transporters typically contain two nucleotide-binding domains and two transmembrane domains (TMD) with each TMD having 6–10 membrane spanning $\alpha$-helices (Neumann et al, 2017). Although ABC transporters can be divided into seven different subfamilies (designated A to G), for our analysis, we distinguished only mitochondrial and non-mitochondrial subgroups of ABC transporters. All genes coding for mitochondrial ABC transporters (B subfamily) were moderately enriched in the MB fraction, with the *ABCB10* showing the smallest enrichment ($\log_2$FC = 0.42) (Fig 3A). The genes encoding non-mitochondrial ABC transporters showed high enrichment in the MB fraction, with the *ABCG5* being the most highly enriched ($\log_2$FC = 2.82) gene. We also observed genes encoding non-mitochondrial SLCs to be more abundant in the MB fraction over HS, whereas genes coding for mitochondrial solute carrier family (SLC25) were enriched in HS fraction (Fig 3B). The SLCs are membrane-bound proteins regulating transport of various substances, including organic cations, ions, and amino acids over the cell membrane. The mitochondrial SLCs transport nutrients required for the energy conversion and maintenance of the cell across the mitochondrial inner membrane (Ruprecht & Kunji, 2020). Each SLCs, in particular a mitochondrial one, has a three-domain structure in which each domain consists of two $\alpha$-helices that are connected by a loop-helix-loop. On the other hand, the MPC comprises MPC1 and MPC2 IMM transmembrane proteins that form a heterodimeric complex enabling pyruvate transport (Bricker et al, 2012). The MPC transports cytosolic pyruvate into mitochondria, where it fuels the tricarboxylic acid (TCA) cycle (Cunningham & Rutter, 2020). Our analysis revealed *mpc1* gene to be more HS-enriched, whereas *mpc2* gene product to localize more to MB fraction (Fig 3C). However, for both of them the reported enrichment is not statistically significant, thus the conclusions regarding genes encoding the MPC components are not credible. Finally, SFXNs are recently discovered five-pass IMM proteins (Tifoun et al, 2021). There are five SFXN genes (*SFXN1-5*) in vertebrates. Although little is known about functions of *SFXN2*, *SFXN4*, and *SFXN5*, the *SFXN1* was convincingly demonstrated to play a role in transport of serine, which is a key metabolic regulator of one carbon metabolism (Kory et al, 2018). The *SFXN3* displays more than 77% amino acid sequence identity and thus is predicted to have a similar role as *SFXN1* (Cunningham & Rutter, 2020). Interestingly, both *sfx1* and *sfxn3* genes appeared to be highly enriched in the MB fraction (Fig 3C), whereas sfxn4 is more HS-enriched. No specific localization changes were observed for *sfxn2* and *sfxn5b* genes. One of the most

---

samples. **(D)** Violin plot showing the distribution of $\log_2$ gene enrichments for genes grouped by their location within mitochondria. **(E)** ER genes were also included in this analysis (E) Bar plots showing the $\log_2$ gene enrichments of yeast (blue) and human (yellow) orthologous genes for which transcripts were reported to be translated by the mitochondrion-bound or free cytosolic polysomes.

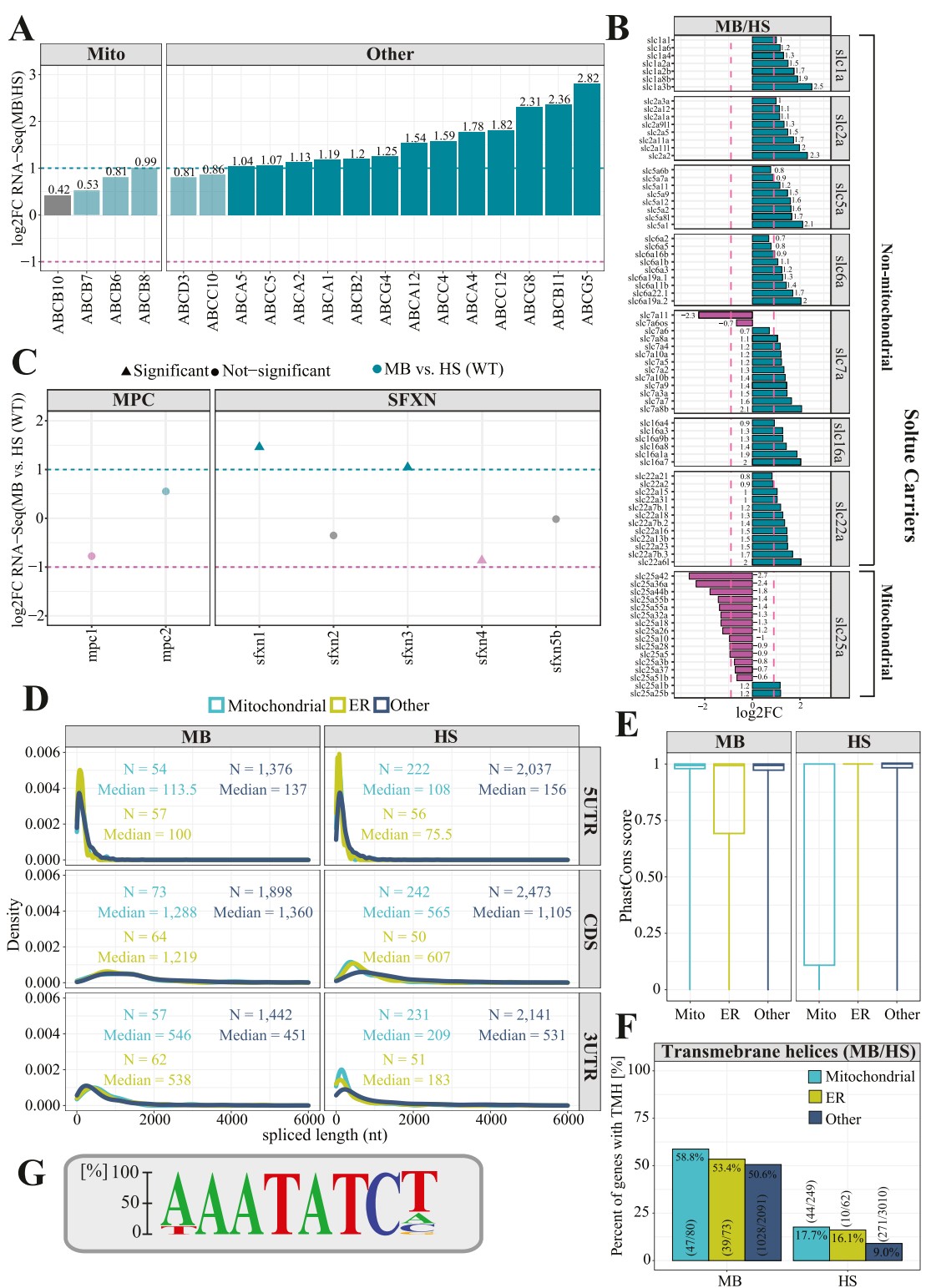

**Figure 3. Properties of membrane-associated transcripts.**
**(A, B, C)** Analysis of gene enrichment of (A) the ATP-binding cassette transporters, (B) solute carriers, (C) mitochondrial pyruvate carrier, and the sideroflexin carriers. The purple dashed line represents log$_2$FC = −1 and the green dashed line represents log$_2$FC = 1. **(D)** Length distribution analysis of coding sequences and UTR regions for transcripts encoding mitochondrial (cyan), ER (green), and other proteins (navy). **(E)** Boxplots showing PhastCons (Siepel et al, 2005) conservation tracks for gene categories described above. PhastCons score equal to 1 represents highly conserved sequences, whereas PhastCons score equal to 0 indicates rapidly evolving sequences. **(F)** Bar plot representing proportion of transcripts encoding mitochondrial, ER and other proteins with transmembrane domains. **(G)** The numbers in

common features among presented transporters is the presence of hydrophobic transmembrane domains. To fulfil their function, proteins with TMDs must protect themselves from random integration into other membranes and aggregation in the cytosol. Therefore, the localized translation coupled with membrane insertion could help to reduce the problem of mis-targeted membrane integration and accumulation of proteins with TMDs in the cytosol. ER can act as the platform for the co-translational import and segregation of proteins with TMDs (Shurtleff et al, 2018). Thus, the enrichment of non-mitochondrial ABC transporters and SLCs in the MB fraction is largely ER driven. In addition, we explored the enrichment of cytochrome P450 enzymes (CYPs). Although CYPs are predominantly localized in the endoplasmic reticulum membranes (ERMs), they also reside in other subcellular compartments, including the plasma membranes and mitochondria (Ahn & Yun, 2010). We explored the expression of CYPs using our data. We were able to detect 42 CYP genes and all of them, except *cyp17a1* were highly associated with membranes. Interestingly, the zebrafish *cyp20a1* gene with yet unknown function that was shown to localize to mitochondria, displayed almost twofold MB-enrichment (Fig S5A).

Finally, we noticed transcripts encoding mitochondrial ribosomal proteins to be entirely enriched in the HS fraction (Fig S5B). This is consistent with previous findings, showing that mitochondrial ribosomal proteins as soluble proteins are mainly synthetized in the cytosol and imported into mitochondrial matrix in their precursor form (Diehn et al, 2000).

We further explored the properties of transcripts enriched in each fraction by comparing their lengths, particularly of their 5' and 3' UTR and coding sequences (CDSs). Surprisingly, CDSs of transcripts encoding mitochondrial proteins that were enriched in MB are two times longer (median = 1,288 nt) compared with the ones enriched in the HS fraction (median = 565 nt, $P$ = 2.51 × 10$^{-11}$, Wilcoxon rank sum test with continuity correction). This is also true to the same extent for ER transcripts (1,219 nt versus 607 nt, $P$ = 3.23 × 10$^{-12}$) and to a lower extent for other transcripts, as MB enriched CDSs are ~30% longer than those for transcripts detected in HS fraction ($P$ = 2.2 × 10$^{-16}$, Wilcoxon rank sum test with continuity correction) (Fig 3D). Moreover, transcripts encoding mitochondrial and ER proteins that have been identified in the MB fraction had on average much longer 3'UTRs (median = 546 and 538 nt) with respect to the HS fraction–enriched transcripts (median = 209 and 183 nt, $P$ = 5.24 × 10$^{-6}$ and $P$ = 2.32 × 10$^{-9}$, respectively, Wilcoxon rank sum test with continuity correction). Whereas the length of 3'UTRs of other transcripts did not vary that much across fractions ($P$ = 8.76 × 10$^{-5}$, Wilcoxon rank sum test with continuity correction). Next, we investigated the evolutionary conservation of mitochondrial, ER, and other transcripts enriched in the MB and HS fractions using a 100-way vertebrate sequence alignment (Blanchette et al, 2004; Siepel et al, 2005). This analysis revealed that the enriched transcripts in the MB fraction, with exception to the ER ones, were highly conserved, whereas the ones present in the HS fraction were more rapidly evolving (Fig 3E). This is particularly true for transcripts

encoding mitochondrial proteins and is consistent with previous findings, indicating that transcripts translated by mitochondrion-bound polysomes are in general much more evolutionary conserved (Marc et al, 2002; Margeot et al, 2005; Garcia et al, 2007). Our results also confirmed that all types (mitochondrial, ER, and other) of membrane associated transcripts were on average likely to encode proteins with transmembrane domains (58.8% versus 17.7%, 53.4% versus 16.1%, and 50.6% versus 9%) (Fig 3F). To investigate whether any other TMD containing proteins are also MB-enriched, we explored the enrichment of the G protein-coupled receptor (GPCR) superfamily that is the largest class of cell membrane receptors (Wess, 1998). All GPCRs contain 7-transmembrane domains (7 TM) of different phylogenetic origin that allows to divide them into nine different subgroups (Schiöth et al, 2010). Of 57 GPCR zebrafish genes identified by Harty et al (2015), only for 10 GPCRs we were able to assign statistically significant enrichment score. All of the detected GPCRs appeared MB-enriched, whereas six of them showed at least twofold MB enrichment (Fig S5C). No information about the subcellular localization was available for any of them.

Finally, to understand the mechanism driving the MB-enrichment, we analyzed the sequence of 3'UTR regions of mitochondrial transcripts detected in the MB fraction using Homer (Heinz et al, 2010) software. This analysis revealed the presence of 16 enriched sequence motifs (Fig S6A). One motif among them – AAAATATCY, where Y in 60% of cases is substituted by T (Fig 3G) was clearly enriched. A motif (TAAATATATAC) (Fig S6B) resembling it was also detected in 3'UTR regions of MB enriched mitochondrial transcripts by the STREME – a recently developed software for motif discovery (Bailey, 2021). Interestingly, among known motifs similar to this one, STREME indicated the Puf3 protein binding motif (TGTAAATA) (Kershaw et al, 2015). Puf3 associates with the cytosolic surface of the OM and guides mRNAs to the mitochondrial surface through the recognition of the binding motif in yeast (Gerber et al, 2004). The motif comparison performed by Tomtom software (Gupta et al, 2007) revealed overlap of six nucleotides between motif sequences (Fig S6B). The FOXJ3—a human transcription factor was another motif detected as similar by the STREME software. These two motifs share eight nucleotides (Fig S6C). The presented motif was detected in 32.2% of analyzed transcript sequences with $P$-value = 7.53 × 10$^{-1}$ and E-value = 2.3, although according to specified settings, STREME should only report the best motifs and those with E-value ≤ 0.05 (Fig S6A).

### Experimental validation of predicted subcellular locations

To validate the computationally predicted enrichment of mitochondrial mRNAs, qRT-PCR experiments using MB and HS fractions were performed. For the validation of MB enriched mitochondrial transcripts (Fig 4A), three candidates from highly enriched category and three from moderately enriched category were selected (Fig 4B). For all selected genes (*comtd1, slc16a1a, cyb5r2, abcb6a, rcn2,* and *chpt1*), we were able to confirm their enrichment in the membrane-bound fraction. As expected, *mt-atp8* encoded by the

parenthesis indicate the numbers used to calculate proportions (G) The sequence logo illustrating motif enriched in 3'UTR regions of MB fraction-enriched transcripts encoding mitochondrial proteins.

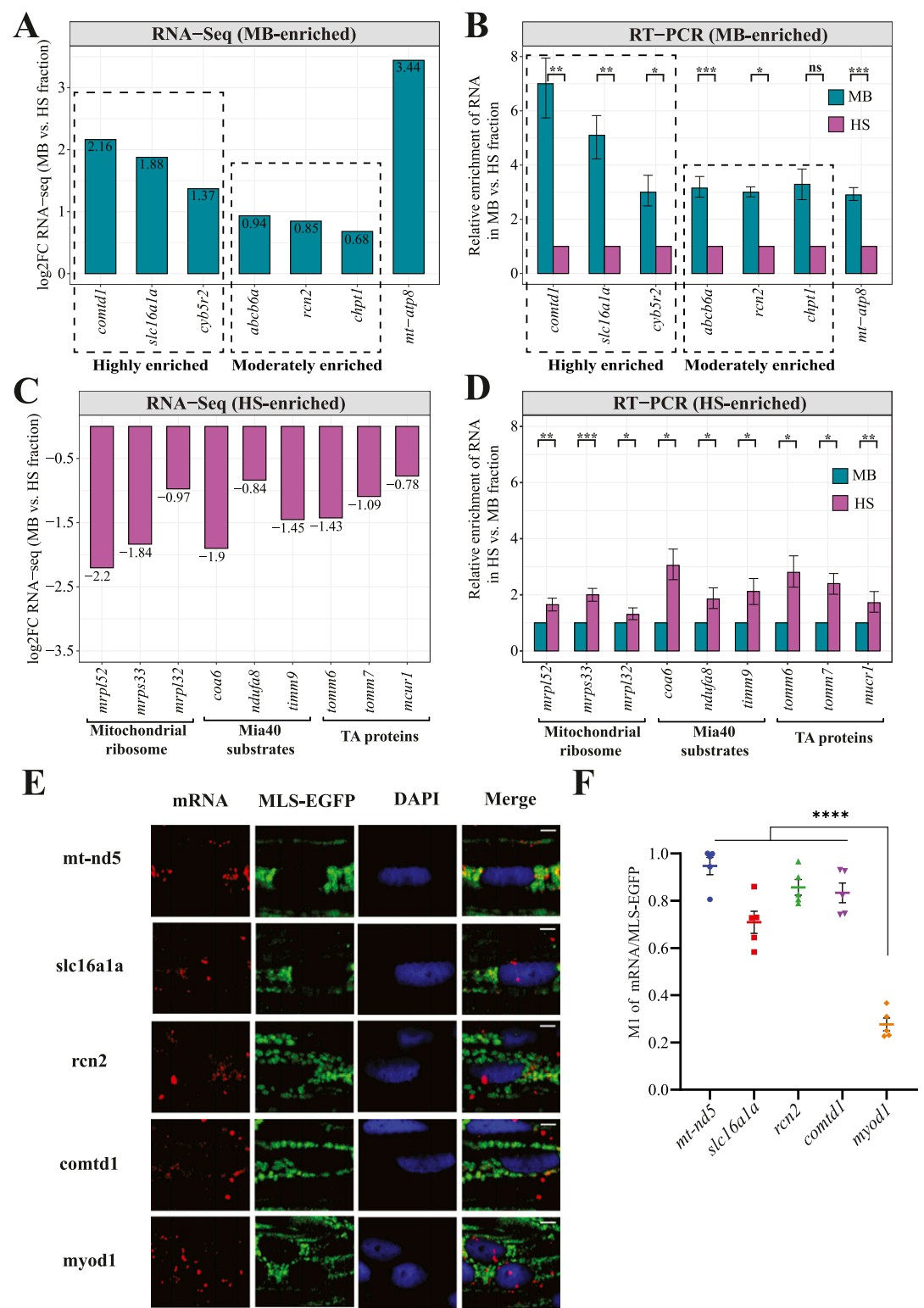

**Figure 4. Experimental validation of computationally predicted gene enrichments.**
**(A)** Bar plot showing the log$_2$ gene enrichments for the MB fraction-enriched MitoCarta 2.0 genes detected as highly and moderately enriched. **(B)** qRT-PCR validation of MB fraction-enriched MitoCarta 2.0 genes. **(C)** Log$_2$ gene enrichments for the HS fraction-enriched MitoCarta 2.0 genes encoding mitochondrial ribosomal proteins, Mia40 substrates and TA proteins. **(D)** qRT-PCR validation of genes enriched in the HS fraction. Relative enrichment of genes in MB with respect to HS upon normalization with ERCC-00096 is shown. Data derived from three biological replicates. Error bars correspond to SEM; $P < 0.05$ (*), $P < 0.01$ (**), $P < 0.001$ (***), and $P < 0.0001$ (****) by unpaired $t$ test. **(E)** Fluorescent in-situ hybridization of selected candidates by RNAscope on whole mount 5 days post fertilization zebrafish larvae. mRNAs are probed by specific RNAscope probes (red) and mitochondria are stained with anti-GFP antibody (green). Yellow in the merged image indicates co-localization of mRNAs on MLS-EGFP

mitochondrial genome has also shown substantial MB enrichment. For the validation of genes enriched in the HS fraction (Fig 4C), we selected candidates encoding different classes of mitochondrial proteins such as Mia40 substrates, tail-anchored (TA) and mitoribosomal proteins. The qRT-PCR results showed that mRNAs encoding mitochondrial ribosomal proteins, Mia40 substrates and TA mitochondrial proteins are enriched in the HS fraction (Fig 4D). Therefore, in all cases, the qRT-PCR results were in the agreement with the RNA-seq data (Fig 4A and C).

To prove that MB enriched mitochondrial transcripts are actually present on the surface of mitochondria in vivo, we performed fluorescent in-situ hybridization by RNAscope technique (Wang et al, 2012a; Gross-Thebing et al, 2014). We selected three MB fraction enriched genes—*rcn2*, *slc16a1a*, and *comtd1*, which are zebrafish orthologues of human genes present in MitoCarta 2.0 gene set (Calvo et al, 2016). The transcripts of *rcn2* and *slc16a1a* have been recently reported by high-throughput APEX-seq study to be associated with OMM in human cells (Fazal et al, 2019). Proteins encoded by these three genes were also shown to be localizing to mitochondria by numerous studies (Brooks et al, 1999; Dubouchaud et al, 2000; Hashimoto et al, 2005, 2006; Hung et al, 2014; Nuebel et al, 2016). As expected, the mtDNA-encoded mRNA, *mt-nd5*, presented the highest co-localization with mitochondria (Fig 4E and F). Transcripts of three genes *rcn2*, *slc16a1a*, and *comtd1* enriched in the MB fraction also displayed high mitochondrial localization. In contrast, *myod1* mRNA, which was not enriched in the MB fraction according to our analysis, presented a cytosolic distribution and the lowest co-localization with mitochondria. Therefore, our RNAscope results indicate that the transcripts of nuclear-encoded mitochondrial genes enriched in MB fraction show high degree of localization on the mitochondrial surface in zebrafish.

## Transcriptome changes triggered by the *chchd4a*$^{-/-}$ mutation

Deficiency in any mitochondrial import machinery results in the accumulation of mitochondrial precursor proteins in the cytosol that triggers the defensive response (Wrobel et al, 2015). Cells protect themselves against defects in mitochondrial biogenesis by both inhibiting protein synthesis and activating the proteasome that performs cellular protein clearance. Because proteasomal activity is regulated by the quantity of mis-localized mitochondria-destined precursor proteins, we hypothesize that defects in the mitochondrial import pathway could alter the localization of mRNAs and translation of nuclear-encoded mitochondrial proteins at the surface of mitochondria to reduce the number of over-accumulated precursor proteins in the cytosol.

To explore this hypothesis, we took advantage of *chchd4a*$^{-/-}$ zebrafish mutant line, in which the import of mitochondrial proteins into IMS was disrupted (Sokol et al, 2018). Mutation in *chchd4a* triggers a glycolytic phenotype in zebrafish, leading to starvation and death starting at 10 days post fertilization (dpf). To examine in details the stress response caused by *chchd4a*$^{-/-}$ mutation, we re-analyzed the RNA sequencing data from our previous study, where we directly compared *chchd4a*$^{-/-}$ to the WT 5 dpf zebrafish samples (Sokol et al, 2018). In total, the differential analysis showed expression changes for 217 genes (FDR 5%; log$_2$FC ± 0.9) with 110 and 107 being up- and down-regulated, respectively (Fig 5A). In general, we observed increased expression of genes involved in amino acid activation and

protein refolding (*hsp70*, *hspd1*, *hspa9*, and *hspa5*) (Fig 5B). Interestingly, we also noticed four genes from the one-carbon pathway (*shmt2*, *mthfd2*, *aldh1l2*, and *mthfd1l*), whose proteins are known to localize to mitochondria, to be up-regulated by *chchd4a*$^{-/-}$ mutation (Fig 5C). We previously observed elevated expression of these genes also at the protein level upon *chchd4a*$^{-/-}$ mutation (Sokol et al, 2018). It has been shown that the one-carbon metabolism can produce mitochondrial NADH under conditions of stress, when the TCA cycle affected by the cell conditions loses its capacity (Ducker & Rabinowitz, 2017). Together with the increased expression of genes encoding mitochondrial and cytoplasmic one-carbon pathway enzymes, we also observed transcriptional activation of serine metabolism, in particular L-serine biosynthetic process (*phgdh* and *psat1*). This is in line with previous observations showing that serine is driving the NADH production via one-carbon metabolism and that various stress responses can increase the transcription of enzymes involved in serine biosynthesis (Zhao et al, 2016; Zhou et al, 2017). Increased transcription of one-carbon metabolism genes suggests a change in the way NADH is produced in the cell. As disruption of mitochondrial functions leads to widespread consequences, we also observed expression changes for genes involved in several metabolic pathways such as proteolysis and cholesterol biosynthetic processes. As previously described *chchd4a*$^{-/-}$ mutation leads to pancreatic insufficiency due to reduced expression of various digestive enzymes and their precursors, which affect the overall development of pancreas and results in inability of *chchd4a*$^{-/-}$ mutants to digest and absorb proteins (Sokol et al, 2018). Whereas, the current analysis revealed an additional link between *chchd4a*$^{-/-}$ mediated mitochondrial insufficiency and the up-regulation of cholesterol biosynthesis. Mitochondrial dysfunction has been previously shown to be associated with increased accumulation of cholesterol in the cell, as mitochondria play a key role in cholesterol metabolism (Martin et al, 2016). Moreover, sterols are crucial for mitochondrial retrograde signalling, which determines the way mitochondria contact with the rest of the cell (Kulig et al, 2016). As a consequence cholesterol synthesis is strictly regulated by the ER, in particular its membrane proteins possessing the ability to sense cholesterol levels (Goldstein et al, 2006). Impairing mitochondrial functions disrupts retrograde signalling, which in turns affects the distribution of free cholesterol in the cell and finally the ability of ER proteins to properly sense sterol levels. Elevated cholesterol levels have been shown to down-regulate mevalonate pathway genes, reducing the biosynthesis of endogenous cholesterol and as a result disrupting the final metabolic outcome of this pathway (Wall et al, 2022). Interestingly, one of the up-regulated genes upon *chchd4a*$^{-/-}$ mutation is *hmgcs1* encoding a key enzyme in the mevalonate pathway. Recently, *HMGCS1*, its human orthologue, has been shown to activate unfolded protein response components (ATF4, XBP1, and ATF6) and protect mitochondria and ER from the damage under proteostatic stress (Zhou et al, 2021). Increased cholesterol biosynthesis can also act as a compensatory mechanism reducing redox stress by consuming elevated NAD(P)H levels in the cell (Schirris et al, 2021).

## Dynamics of transcript subcellular localization under mitochondrial stress conditions

To explore the impact of *chchd4a*$^{-/-}$ mutation on the subcellular localization of nuclear-encoded mitochondrial transcripts, we

obtained MB and HS fractions via biochemical fractionation from 5 dpf *chchd4a*$^{-/-}$ zebrafish larvae (Fig S7). Next, we isolated poly-adenylated RNA from each fraction and performed RNA-seq. We followed the same data analysis steps, in which we directly compared the MB with the HS and assigned genes to each fraction based on their enrichment (FDR 5%; log$_2$FC <=> ±1). Again, we noticed two populations of genes either enriched in the MB or HS fraction (Fig S8A). No major changes in the number of detected genes in *chchd4a*$^{-/-}$ compared with WT samples were observed, except for the MB fraction, in which the number of enriched genes increased by almost a 1,000 (Fig S8B). We also noticed that the *chchd4a*$^{-/-}$ mutation affected the enrichment of mitochondrially encoded genes in the MB fraction, particularly for three genes: *mt-nd3*, *mt-nd6*, and *mt-atp6* (Fig S8C). To determine the function of genes identified in each fraction, we performed a GO and KEGG enrichment analysis. We did not notice any specific changes in the GO enrichment analysis in either "biological process" or "cellular component" searches (Fig S8D and E). However, the KEGG analysis revealed further enrichment of genes involved in the OXPHOS, together with genes encoding cytosolic and mitochondrial ribosomal proteins in the HS fraction compared with WT (Fig S8F). In contrast, genes detected in the MB fraction were mainly enriched in general KEGG terms, including lysosome, sphingolipid metabolism, or ECM–receptor interaction. Similar to the analysis strategy for the WT, we again tested our gene lists against a merged set of zebrafish orthologues for genes in MitoCarta 2.0 (Calvo et al, 2016) and IMPI inventories. Indeed, we observed an increased presence of nuclear-encoded mitochondrial genes in the HS fraction for both highly and moderately enriched categories, whereas in the MB, there is a slight increase in the number of mitochondrial transcripts in the highly enriched category of *chchd4a–/–* mutants as compared with WT (Fig 6A). This was also clearly seen in the distribution of enrichment values for specific mitochondrial compartments (Fig 6B) by a noticeable shift towards HS fraction for genes from all the mitochondrial sub-localizations. To understand if this shift was directly caused by the *chchd4a*$^{-/-}$ mutation, we investigated the enrichment of MIA pathway targets. This analysis confirmed that the *chchd4a*$^{-/-}$ mutation, indeed, enhanced the HS fraction enrichment for almost all of analyzed MIA pathway targets (Fig 6C).

To better understand the effect of *chchd4a*$^{-/-}$ mutation on the subcellular localization of mitochondrial gene products we tested our enrichment scores against the mitochondrial enrichment in yeast reported by the Weissman group (Williams et al, 2014). We again observed poor correlation between the MB enrichment in zebrafish and the OM45 enrichment in yeast with and without CHX (Fig S9A and B). Similarly to WT samples, we observed high enrichment of genes encoding two ABC transporters *abcb11a* and *abcb11b* (with homology to *MDL1* and *MDL2* in yeast). Although we detected new *MDL1* and *MDL2* zebrafish orthologues—*abcb4* and *abcb6a* to be MB-enriched upon *chchd4a*$^{-/-}$ mutation, we did not identify *tap1*—zebrafish orthologue that was detected in the WT MB fraction (Fig S9C). Moreover, we noticed further MB enrichment of

SLCs including *slc30a2* and *slc25a35*, which are zebrafish orthologues of *MMT1*, *MMT2*, and *OAC1* yeast genes, respectively. In addition, *chchd4a*$^{-/-}$ mutation increased the MB-enrichment of CU856539.1—zebrafish orthologue of ARN2 yeast gene that encodes transporter specifically recognizing siderophore-iron chelates. These genes were detected in both yeast samples treated and untreated with CHX (Fig S9C and D). However, similarly to the analysis for the WT fractions, the overlap between MB-enriched genes and CHX treated yeast samples was lower compared with the CHX-untreated ones.

To further investigate if mitochondrial protein import disorders had affected the properties of membrane-associated transcripts, we compared the transcript lengths between the MB and HS fractions in *chchd4a*$^{-/-}$. Interestingly, mitochondrial, ER, and other transcripts enriched in the MB fraction have on average even longer CDSs (Fig S10A), when compared with the WT MB fraction (Fig 3D). The median length increased by ~200 for mitochondrial and ER transcripts (*P* = 2.2 × 10$^{-16}$ and *P* = 3.67 × 10$^{-11}$ according to Wilcoxon rank sum test with continuity correction), whereas for ~300 nucleotides for other types of transcripts (*P* = 2.2 × 10$^{-15}$, Wilcoxon rank sum test with continuity correction). Moreover, we noticed that the mitochondrial transcripts in the MB fraction of *chchd4a*$^{-/-}$ mutants had shorter 3′UTR regions (median = 424.5 nt) than those identified in WT MB fraction (median = 546, *P* = 0.00017 according to Wilcoxon rank sum test with continuity correction). This effect is less pronounced for the ER transcripts (median = 538 nt versus median = 496 nt). Surprisingly, the length of CDSs in other types of transcripts enriched in the HS fraction was reduced by ~40% (1,105 versus 667, *P* = 0.00023). This means that the average CDS length of transcripts enriched in the MB fraction are almost three times longer than those detected in the HS fraction. Thus, the difference between the CDS length for transcripts in MB and HS fractions and hence, the size of the protein, was even larger for *chchd4a*$^{-/-}$ mutants compared with the WT. Surprisingly, similar to that in WT, transcripts enriched in the MB fraction of *chchd4a*$^{-/-}$ mutant were also more evolutionarily conserved than those in the HS fraction according to the analysis performed using 100-way Multiz vertebrate alignment (Blanchette et al, 2004; Siepel et al, 2005) (Fig S10B). Interestingly, we observed the biggest change in the evolutionary conservation scores for the mitochondrial transcripts. At the same time, *chchd4a*$^{-/-}$ mutation reduced the presence of transcripts encoding proteins with TMDs in the MB fraction for both mitochondrial and other transcripts, whereas this proportion remained similar for ER transcripts (Fig S10C). Finally, we saw a slight decline in MB-enrichment for genes encoding mitochondrial ABC transporters upon *chchd4a*$^{-/-}$ mutation (Fig S10D).

To further explore the relationship between the protein size and the membrane association, we plotted the CDS length versus the enrichment score for both WT and *chchd4a*$^{-/-}$ fractions. We did not observe major differences for WT MB and HS fractions (Fig S11A) when looking at all genes. Whereas, there was a large shift in the CDS length for *chchd4a*$^{-/-}$ samples, as MB-enriched genes had on average much longer CDSs than the ones enriched in the HS

tagged mitochondria. Scale = 5 µm. **(F)** Analysis of co-localization of MB fraction-enriched MitoCarta 2.0 mRNAs with mitochondria. M1 indicates Mander's co-localization co-efficient of mRNA fraction co-localized on mitochondria harboring MLS-EGFP. The difference between *myod1* negative control and mRNA candidates were analyzed by unpaired *t* test. Error bars correspond to SEM and data derived from five region of interests originating from three experiments (*n* = 5); *P* < 0.0001 (****).

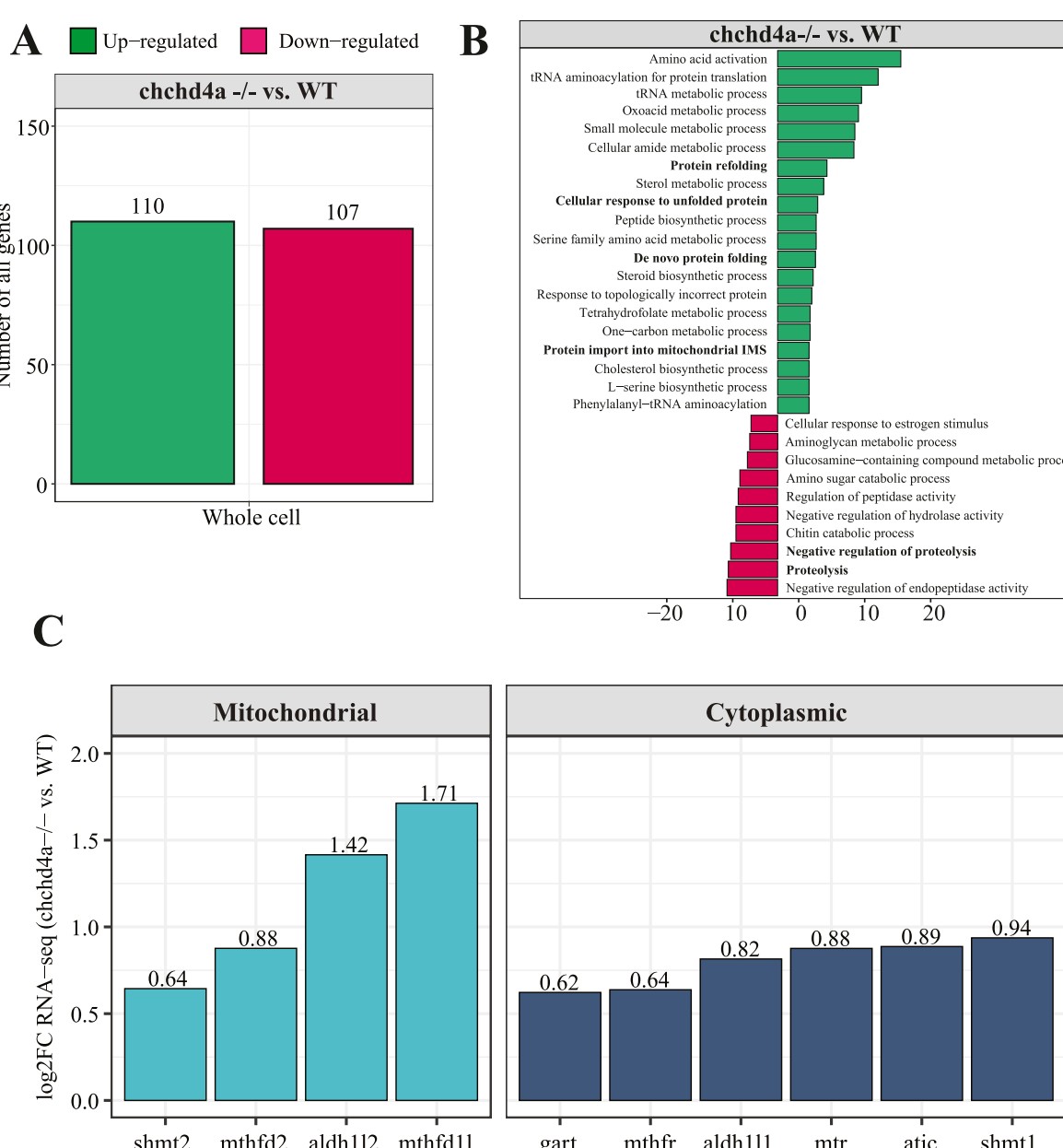

**Figure 5. Transcriptome changes triggered by *chchd4a*$^{-/-}$ mutation.**
**(A)** Numbers of differentially expressed genes between whole unfractionated *chchd4a*$^{-/-}$ and WT samples. **(B)** KEGG enrichment analysis for differentially expressed genes for the whole unfractionated samples. **(C)** Bar plot showing expression changes of one-carbon genes encoding cytoplasmic (dark blue) and mitochondrial enzymes (light blue).

fraction (Fig S11B). However, the values of CDS length increased gradually and it was not possible to designate a specific cut-off point that could help to distinguish the MB-enriched from HS-enriched genes solely based on the size of the encoded protein. This trend was also visible when looking only at mitochondrial genes (Fig S11C and D). Interestingly, it has been shown that there is a relationship between the protein conservation and sequence length, as highly conserved proteins have on average longer sequences (Lipman et al, 2002). To investigate whether the sequence conservation could in any way affect the sequence length, we

examined the length distributions for lowly (<0–0.25), medium (<0.25–0.75), and highly (<0.75–1) conserved CDS sequences (Fig S11E). Interestingly, more conserved CDS sequences were on average longer than the poorly conserved ones (median = 1,669 nt versus median = 1,278 nt, $P = 1.2 \times 10^{-10}$ according to Wilcoxon rank sum test with continuity correction). This effect was even more pronounced for very well annotated genomes such as the human genome (Fig S11F, median = 5,024 nt versus median = 12,971 nt, $P = 3.4 \times 10^{-8}$ according to Wilcoxon rank sum test with continuity correction). Therefore, one of the important aspects contributing

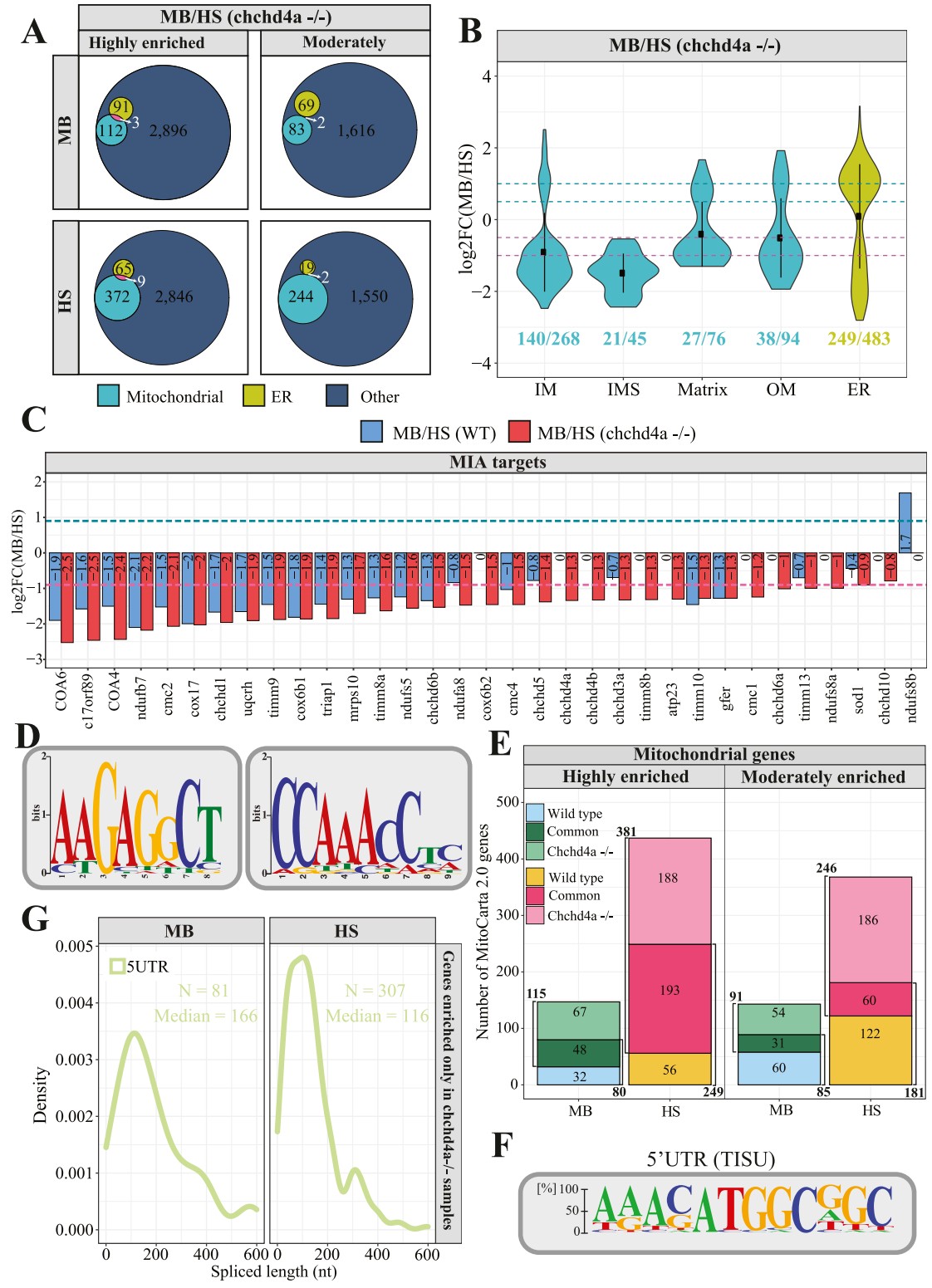

**Figure 6.   Dynamics of mRNA subcellular localization under mitochondrial protein import deficiency.**
**(A)** Venn diagrams representing the number of mitochondrial genes from the merged MitoCarta 2.0 (Calvo et al, 2016) and IMPI repository, as well as the zebrafish orthologues of human 483 genes encoding proteins that localize to ER according to the Human Protein Atlas in each *chchd4a⁻ᐟ⁻* set. **(B, C)** Violin plot showing the distribution of log₂ *chchd4a⁻ᐟ⁻* gene enrichments for genes grouped by their location within mitochondria, including also ER genes; (C) gene enrichment analysis for MIA substrates in transcriptomic data for WT (blue) and *chchd4a⁻ᐟ⁻* (red) 5 days post fertilization samples. The purple dashed line represents log₂FC = −1 and the green dashed line represents log₂FC = 1. **(D)** Two sequence logo illustrating motifs detected in 3'UTR regions of MB fraction-enriched transcripts encoding mitochondrial

to the differences in the CDS length encoded by the genes enriched in the MB fraction can be the sequence conservation.

Despite the reduced length of 3'UTRs of mitochondrial transcripts enriched in the MB fraction upon $chchd4a^{-/-}$ mutation, using Homer (Heinz et al, 2010) software we confirmed the presence of previously detected sequence motif (AAATATCY, 26.69%, $P$-value = $1 \times 10^{-2}$) (Fig 3G) among 15 detected motifs (Fig S12A). However, we were not able to detect this motif using STREME (Bailey, 2021) software (Fig S12B). Instead, STREME reported two new motifs enriched in 3'UTR regions of mitochondrial transcripts detected in the MB $chchd4a^{-/-}$ fraction – AAGAGCCT and CCAAACCTC (Fig 6D). These motifs were found in 26.6% ($P$-value = $5.0 \times 10^{-2}$) and 35.4% ($P$-value = $5.0 \times 10^{-2}$) 3'UTR sequences of mitochondrial transcripts, respectively. Although the mechanisms driving the localization of mitochondrial transcripts to the MB fraction upon $chchd4a^{-/-}$ mutation remain unknown, our data suggest that the $chchd4a$ loss of function further limits the type of localized mRNAs to well-conserved transcripts encoding large proteins.

To explore the increased enrichment of mitochondrial transcripts in the HS fraction, we further investigated the overlap between genes enriched in each fraction for WT and $chchd4a^{-/-}$. Our results indicate that 60% (48) and ~80% (193) of highly enriched genes are shared between $chchd4a^{-/-}$ and WT MB and HS fractions, respectively (Fig 6E). This could suggest that the $chchd4a^{-/-}$ mutation is further enriching the RNA subpopulations in our fractions, rather than substantially changing them. This statement is particularly true for the highly enriched genes in the HS fraction. However, we observed only ~30% of overlap between moderately enriched genes in the WT and $chchd4a^{-/-}$ HS and MB fractions, respectively. At the same time, the $chchd4a$ loss of function triggers exclusive enrichment of 67 and 188 highly genes in the MB and HS fractions, respectively. As indicated by the KEGG pathway enrichment, $chchd4a^{-/-}$ seems to affect the energy metabolism (Fig S13A). Genes detected in the HS fraction of $chchd4a^{-/-}$ mutants were directly involved in the OXPHOS and citrate cycle. At the same time, we observed a handful non-mitochondrial genes (Fig S13B) and no mitochondrial genes (Fig S13C) switching the enrichment between WT and $chchd4a^{-/-}$ mutant MB and HS fractions.

Analysis of the 5'UTR regions of genes exclusively enriched in the HS upon $chchd4a$ loss revealed a sequence element that might have increased their presence there. The translation initiator of short 5' UTR (TISU) element (Elfakess & Dikstein, 2008; Elfakess et al, 2011) is known to regulate translation of mitochondrial genes, as it confers resistance to translation inhibition in response to energy stress (Sinvani et al, 2015; Gandin et al, 2016). Therefore, we suspect that TISU elements may play a regulatory role in the translation of mitochondrial genes upon $chchd4a$ loss of function. As expected, we found the TISU element (SAAS**AUG**GCGGC) highly enriched among the 5'UTR sequences of genes exclusively enriched in the HS fraction (E-value = $2.1 \times 10^{-14}$) of the $chchd4a^{-/-}$ mutants (Fig 6F). We also noticed that in $chchd4a^{-/-}$ mutants, the 5'UTRs of mitochondrial transcripts exclusively enriched in the HS were much

shorter compared with those enriched in MB fraction (median = 106 nt versus median = 166 nt) (Fig 6G). Interestingly, transcripts exclusively enriched in the MB fraction upon $chchd4a^{-/-}$ mutation had almost three times longer CDS sequences compared with transcripts enriched in the HS (Fig S13D). We did not detect any specific sequence motif enriched among the 5'UTRs of genes exclusively enriched in the MB fraction.

# Discussion

Here, we present the genome-wide analysis, which provides new insights into subcellular localization of nuclear-encoded mitochondrial gene products in zebrafish. In this study, we developed a biochemical fractionation method that allows to obtain the membrane-bound fraction containing intact mitochondria and ER with ribosomes on their surface. We showed that this strategy enables to accurately analyze mitochondrial transcripts that are likely to be translated by the free cytosolic and membrane-associated ribosomes. By biochemically fractionating zebrafish 5 dpf samples and measuring the RNA abundances in each fraction, we captured the subcellular landscape of nuclear-encoded mitochondrial mRNAs. We also detected various sequence features that may determine the subcellular localization of different mitochondrial transcripts. Moreover, in this study, we describe consequences of proteostatic stress response triggered by the disorders in the mitochondrial protein import for the localization of nuclear-encoded mitochondrial transcripts.

Recent technological advancements in the RNA sequencing field also triggered a substantial progress in studying RNA at subcellular resolution. Currently, there are two main methods for producing quantitative subcellular RNA maps: (1) "CeFra-seq" developed within the ENCODE project (Djebali et al, 2012), where the cells are fractionated to extract RNA from specific compartments (Lefebvre et al, 2017; Benoit Bouvrette et al, 2018) and (2) the APEX-seq method, using transgenic labelled proteins of known localization to cross-link nearby RNAs (Fazal et al, 2019). Although both methods are very versatile and can capture the landscape of all RNA types (coding and noncoding) for any subcellular region in the cell, the application of APEX-seq in vivo is more challenging. This method uses APEX2 peroxidase that is genetically targeted to the cellular region of interest to tag endogenous RNAs using proximity biotinylation. This requires genetic manipulations that allow incorporation of these elements into the genome of studied organism. Although similar genetic modifications have been previously described for zebrafish, in general production of transgenic lines for vertebrate species is not only much more challenging and time consuming, but also often requires specialized equipment (Simões et al, 2020). New genetic modifications are particularly problematic for mutant lines, especially if the mutation inhibits the development or turns out to be lethal in its early stage, as in the case of the $chchd4a^{-/-}$ mutation in zebrafish. Another possibility is a proximity

proteins in $chchd4a^{-/-}$ samples identified by STREME software (Bailey, 2021). **(E)** Venn diagrams depicting the overlap between the genes identified in WT and $chchd4a^{-/-}$ MB and HS fractions, respectively. **(F)** The sequence logo illustrating TISU element found in the 5'UTR regions of transcripts exclusively enriched in the HS upon $chchd4a$ mutation. **(G)** Distribution of 5'UTR lengths in which TISU element was detected.

specific ribosome profiling that enables investigation of localized translation genome wide at subcellular resolution. It has been mainly used to profile in vivo actively translated mRNAs on the ERM (Jan et al, 2014) and on the surface of mitochondria (Williams et al, 2014). However, this method uses a spatially restricted biotin ligase (BirA) to label ribosomes with a biotin acceptor peptide (AviTag) in live cells. Therefore, limitations of adopting this strategy for zebrafish are similar as for the APEX-seq approach (Fazal et al, 2019). Finally, other methods for direct analysis of RNA localization at high-throughput exist, including highly multiplex RNA profiling. Nonetheless, these methods require designing thousands of bar-coded oligonucleotide probes to target RNA of interest (Chen et al, 2015). Moreover, they can only measure a limited number of RNA molecule types in a single experiment, which often does not exceed 1,000 RNA types. One of the major drawbacks of all fluorescence in-situ hybridization based approaches is the need for cell fixation and permeabilization that can lead to the displacement of cellular components (Schnell et al, 2012). Considering the above, biochemical fractionation is still one of the most accessible approaches for studying the subcellular localization of nuclear-encoded mito-chondrial gene products in zebrafish, in particular under perturbed conditions.

Many studies reported the presence of mRNAs on the mito-chondrial surface in yeast (Suissa & Schatz, 1982; Egea et al, 1997; Corral-Debrinski et al, 2000; Diehn et al, 2000; Marc et al, 2002; Gadir et al, 2011; Lesnik et al, 2014, 2015; Williams et al, 2014; Tsuboi et al, 2020) and in vitro human cell lines (Matsumoto et al, 2012; Gehrke et al, 2015; Fazal et al, 2019), but so far no such analysis has been performed in vivo in higher eukaryotes. Here, we investigated the population of mRNAs encoding mitochondrial proteins in the mi-tochondrial vicinity in vivo using zebrafish. Our results indicate that the vast majority of mitochondrial transcripts were likely to be translated by the free cytosolic polysomes, whereas just a small fraction of them was detected in the membrane-bound fraction. Interestingly, compared with yeast (Diehn et al, 2000; Buxbaum et al, 2015; Avendaño-Monsalve et al, 2020), the population of mito-chondrial transcripts located in the proximity of the mitochondrial surface in zebrafish was much more modest. We could not confirm the presence in the membrane-bound fraction for the majority of mitochondria enriched transcripts that were reported for yeast (Figs 2E and S4A and B). On the other hand, we observed the en-richment of zebrafish orthologues (abcb11a, abcb11b, and tap1) of MDL1 and MDL2 genes encoding ABC transporters in yeast that were also shown to be present on the surface of mitochondria in yeast (Williams et al, 2014). Interestingly, we also noticed zebrafish orthologues (slc30a1a and slc30a8) of MMT1 and MMT2 – SLCs known to be translated on the mitochondrial surface to be highly enriched in the MB fraction (Fig S4C and D). Moreover, the vast majority of mRNAs reported to be located in the proximity of mi-tochondrial surface in human cell lines (Fazal et al, 2019) were also enriched in our MB fraction, including the zebrafish orthologue of human SLC SLC39A10. Our data suggest that, in vertebrates, mito-chondrial membrane associated transcripts mainly consisted of mRNAs with specific properties, including long CDSs and long 3′UTRs (Fig 3D). In addition, we identified three new sequence motifs that are enriched in 3′UTR regions of membrane associated mRNAs encoding mitochondrial proteins in WT and $chchd4a^{-/-}$

samples, respectively (Figs 3G and S6D). This opens new questions on the factors and exact mechanism involved in mRNA targeting to mitochondrial membrane.

Our results suggest that the post-translational import pathway is likely the main route for mitochondrial proteins, whereas the co-translational import is rather a specialized path intended mainly for transport of large proteins with particular properties that could make the import of nascent protein chains inefficient or prob-lematic for the cell. The observation that genes encoding IMS proteins were mainly enriched in the HS fraction suggests that IMS proteins are likely to be translated by free cytosolic polysomes and imported into IMS after completing their synthesis in the cytosol. This is in line with expectations, as the Mia40-mediated import of IMS proteins import involves oxidation-mediated compaction and requires those proteins to be of very small size (Backes & Herrmann, 2017). On the other hand, we noticed that large pro-teins with highly hydrophobic transmembrane domains, including ABC transporters, SFXNs, and SLCs, were enriched in the membrane-bound fraction.

We also observed the enrichment of genes encoding other TMD containing proteins—GPCR that contain 7-transmembrane domains (7 TMD) (Wess, 1998). Thus, describing any potential link between function of these proteins and the enrichment of their transcripts in the membrane-bound fraction requires additional analysis that perhaps would also reveal the enrichment of genes encoding other classes of proteins with TMDs in the MB fraction. So far, we can only conclude that directly coupling translation with protein import can protect the cell from the toxic effects of protein accumulation or erroneous incorporation of those proteins into other membranes. Therefore, this suggests that the protein properties encoded in the primary transcript sequence might determine the choice of import route already at the mRNA stage. Another important feature of transcripts associated with membranes is their high evolutionary conservation, whereas transcripts translated by the free cytosolic polysomes are much more rapidly evolving (Fig 3E). This finding is in line with previous reports and might be one of the strategies ensuring the fidelity during mitochondrial protein import by en-abling only the co-translational uptake of stable, core mitochon-drial proteins (Fox, 2012).

Our RNA-seq based findings are supported by the experimental validation using both RT-PCR and fluorescent in-situ hybridization by RNAscope technique (Wang et al, 2012a; Gross-Thebing et al, 2014) (Fig 4). Interestingly, we noticed the enrichment of slc16a1a gene in the MB fraction. The RNAscope together with RT-PCR results confirmed the mitochondrial localization of this gene product. The SLC16A1 (MCT1) is a proton-dependent monocarboxylate trans-porter, involved in transportation of lactate and pyruvate, across the plasma membrane. Although the SLC16A1 was initially anno-tated as a mitochondrial protein according to the MitoCarta 2.0 gene set (Calvo et al, 2016), as a plasma membrane protein, it was excluded from its updated version of the MitoCarta 3.0 (Rath et al, 2021). Despite the absence of this protein in the current MitoCarta 3.0 database, there is evidence for localization of SLC16A1 in mito-chondria. For example, Brooks's laboratory showed that in rat skeletal muscle (Hashimoto et al, 2006) and neurons of rat brain regions (Hashimoto et al, 2008), SLC16A1 can be expressed both in plasma and mitochondrial membranes. This study indicates that

SLC16A1 can facilitate both intracellular and cell-cell lactate shuttles and together with mitochondrial lactate dehydrogenase can be involved in lactate oxidation. It is worth to mention that above results solely refer to the SLC16A1 localization at the protein level, whereas in our study we study RNA localization. Interestingly, APEX-seq data revealed that *SLC16A1* mRNA localize to specific subcellular compartments, mainly ERM and OMM (Fazal et al, 2019). Dual subcellular localization of *SLC16A1* transcripts can be explained by the large interactions and physical contacts between ER and mitochondria in the cell. Nevertheless, the full understanding of mechanisms driving co-localization of *SLC16A1* transcript with the mitochondrial membrane requires further investigation.

Biogenesis of fully functional mitochondria depends on a balance between synthesis and degradation of many cellular proteins (Topf et al, 2019). To understand whether stress conditions can increase the interaction between two types of import, we investigated the dynamics of subcellular localization of mRNAs encoding mitochondrial proteins using *chchd4a*$^{-/-}$ zebrafish mutant line (Sokol et al, 2018). Surprisingly, disorders in mitochondrial IMS protein import triggered by the disruption of MIA pathway further restricted the membrane associated population of mRNAs to mainly transcripts with the longest coding (Fig S10A) and the most evolutionarily conserved sequences (Fig S10B). At the same time, we observed the reduction of the length of CDSs for transcripts enriched in the HS fraction upon the loss of *chchd4a* function. The MB and HS enriched transcripts populations differed with respect to their CDS length. The CDSs of transcripts enriched in the MB were three times longer as for the ones enriched in HS fraction (Fig S10A). Our analysis revealed a large shift in the CDS length for transcripts enriched in the *chchd4a*$^{-/-}$ samples. Transcripts enriched in the MB *chchd4a*$^{-/-}$ fraction are on average much longer than for the HS (Fig S11B). The Weissman group showed that inhibiting the translation elongation with CHX gives more time for ribosome–nascent chains to engage mitochondria, as a result they observed a substantial increase in the number of mitochondria enriched proteins (68%) in CHX-treated compared with CHX-untreated samples (27%) (Williams et al, 2014). A longer CDS sequence requires more time for protein synthesis, which in turn gives more time for a presequence to bind the TOM complex. Thus, the enrichment of transcripts with longer sequences in the MB fraction upon *chchd4a*$^{-/-}$ mutation can be related to the kinetics, especially that we observe a gradual increase in the CDS length versus the MB enrichment. However, it has been also shown that highly conserved proteins have on average longer sequences (Lipman et al, 2002). Our analysis confirmed this finding, as more conserved CDS sequences were on average longer than the poorly conserved ones (median = 1,669 nt versus median = 1,278 nt, $P$ = 1.2 × 10$^{-10}$ according to Wilcoxon rank sum test with continuity correction) (Fig S11E). The observed difference is much larger for very well annotated genomes such as the human genome (median = 5,024 nt versus median = 12,971 nt, $P$ = 3.4 × 10$^{-8}$ according to Wilcoxon rank sum test with continuity correction) (Fig S11F). Although the full understanding why the longer proteins are more likely to be co-transcriptionally imported requires further investigation, it could be though that the link between the protein size, protein conservation and prolonged time of the synthesis is an important factor determining the protein import route. We also noticed that *chchd4a* loss of function additionally increased the HS

enrichment of MIA pathway targets that are known to be small-sized proteins (Fig 5C). Interestingly, it has been recently described that mRNA association with mitochondria differs between fermentative and respiratory conditions in yeast (Tsuboi et al, 2020). A switch to the respiratory conditions enhances the localization of certain nuclear-encoded mRNAs to the surface of mitochondria and helps to increase the protein synthesis thorough mRNA localization.

Our results indicate that 60% (48) and almost 80% (193) genes found in MB and HS fractions, respectively, are present in both WT and *chchd4a*$^{-/-}$ samples. This suggests that the stress response triggered by the disruption of mitochondrial import in *chchd4a*$^{-/-}$ mutants further enhanced the presence of specific transcript species, rather than changing it completely. This statement is particularly true for HS fraction. Detailed analysis revealed that many transcripts exclusively enriched in *chchd4a*$^{-/-}$ mutants were involved in energy metabolism (Fig S13A) and their increased abundance in the HS fraction coincide with the presence of the Translation Initiator of Short 5′ UTR (Elfakess & Dikstein, 2008; Elfakess et al, 2011) (TISU) element in the 5′UTR regions of their transcripts. TISU confers resistance to translation inhibition in response to energy stress and is known to regulate the translation of mitochondrial genes (Sinvani et al, 2015; Gandin et al, 2016). Moreover, a recent study investigating the cellular temporal proteostatic stress response in human cell lines, reported proteins localized to mitochondria as the only category of genes significantly up-regulated in translation in response to ER stress (Rendleman et al, 2018). The authors showed that the examination of the 5′ UTRs revealed the presence of TISU elements in Complex I–IV mitochondrial genes up-regulated by the ER stress response.

The *chchd4a*$^{-/-}$ mutation triggers similar transcriptomic changes as the tunicamycin induced ER stress (Rendleman et al, 2018). In both cases, cells struggle with the accumulation of misfolded proteins that in the case of the ER stress is repressing translation of TCA cycle genes. We could not confirm reduced levels of TCA cycle enzymes at the protein level using our mass spectroscopy data from previous study (Sokol et al, 2018). Moreover, we observed slightly increased (log$_2$FC < 0.5) expression of some TCA cycle genes (data not shown). At the same time, we noticed increased activity of one-carbon metabolism (Fig 5B and C) and serine biosynthesis, which can suggest that cells are trying to use this alternative route for the NADH production. We also noted that *chchd4a*$^{-/-}$ mutation enhances the amino acid activation. This is in line with previous findings reporting this up-regulation for the ER stress response, so far without any possible explanation (Ventoso et al, 2012; Cheng et al, 2016).

Taken together, this study provides new insights into mitochondrial mRNA localization under physiological and pathological conditions resulting from disruption of mitochondrial import pathway. Contrary to our initial hypothesis and recent reports in yeast (Williams et al, 2014; Tsuboi et al, 2020), in vivo analysis in zebrafish revealed that localized translation is not the predominantly used mechanism for the synthesis of mitochondrial proteins. It is rather a specialized path intended to import large and well conserved proteins with special properties like presence of TMD (Fig 7). Interestingly, proteostatic stress response further reduces the capacity of the co-translational import by limiting it only to the longest and the most conserved proteins. In general, the protein size seems to be an important factor for the protein synthesis and import, especially under proteostatic

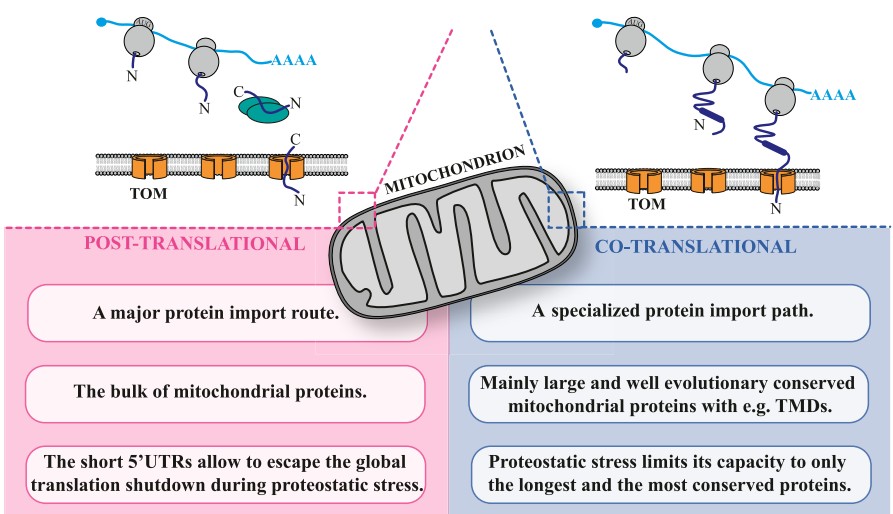

Figure 7.  A model describing principles of mitochondrial protein import in zebrafish.

stress, as short proteins with short 5'UTRs can escape the global translation shutdown triggered by *chchd4a*$^{-/-}$ mutation.

# Materials and Methods

### Zebrafish lines and maintenance

Zebrafish WT line from AB background and *chchd4a*$^{+/-}$ mutant line (Sokol et al, 2018) in the *Tg(Xla.Eef1a1:mlsEGFP)cms1* transgenic background (Kim et al, 2008a, 2008b) were maintained in the IIMCB zebrafish core facility (License no. PL14656251) in accordance with institutional and national ethical and animal welfare guidelines. The optimization of biochemical fractionation was carried out using the AB line. 5 dpf larvae obtained from an in-cross of *chchd4a*$^{+/+}$ siblings in homozygous *Tg(Xla.Eef1a1:mlsEGFP)cms1* background were used as the WT reference for fractionation, RNA sequencing and qRT-PCR validations. Embryos obtained from the in-cross of heterozygous *chchd4a*$^{+/-}$ in homozygous *Tg(Xla.Eef1a1:mlsEGFP)cms1* background were observed under a fluorescent microscope at 4 dpf to select *chchd4a*$^{-/-}$ individuals based on the characteristic GFP-containing abnormal mitochondrial structures described previously (Sokol et al, 2018). The larvae of *chchd4a*$^{+/+}$ and *chchd4a*$^{-/-}$ in *Tg(Xla.Eef1a1:mlsEGFP)cms1* background were grown in an incubator at 28°C until 5 dpf and used for fractionation-sequencing.

### Subcellular fractionation of 5 dpf larvae and isolation of membrane-bound and high-speed fractions

All steps were carried out at 4°C unless otherwise specified. For subcellular fractionation, around 250 zebrafish larvae at 5 dpf were anesthetized using Tricaine (A5040; Sigma-Aldrich) in E3 medium, washed thrice with Ringer's solution (116 mM NaCl, 2.9 mM KCl, and 5.0 mM HEPES, pH 7.6) and incubated in cold isolation buffer (IB) composed of 220 mM mannitol, 70 mM sucrose, and 20 mM HEPES (pH 7.6), for at least 30 min. The larvae were suspended in 4 ml of the IB with EDTA (IB-E; 20 mM HEPES [pH 7.6], 220 mM mannitol, 70 mM

sucrose, and 1 mM EDTA, supplemented freshly with 2 mM PMSF) or in IB with magnesium (IB-M; 20 mM HEPES [pH 7.6], 220 mM mannitol, 70 mM sucrose, 5 mM MgCl$_2$, supplemented freshly with 2 mM PMSF, 200 $\mu$g/ml CHX, and 10 U/ml RiboLock RNase inhibitor). Then the larvae were homogenized in a 5-ml Dounce glass homogenizer equipped with a tight-fitting glass pestle by giving 15 strokes. The homogenate (total) was centrifuged twice at 1,500$g$ for 10 min to remove nuclei and debris to obtain the post-nuclear supernatant. The post-nuclear supernatant was further centrifuged at 14,000$g$ for 15 min to pellet the membrane-bound (MB) fraction that is expected to contain mitochondria. The resulting supernatant was centrifuged at 100,000$g$ for 60 min to obtain the second pellet—the high-speed (HS) and the cytosol as supernatant.

Isolation of crude and EDTA-stripped MB and HS fractions from WT and *chchd4a*$^{-/-}$ were performed as described above, with the following modification. The IB-M and IB-E buffers were additionally supplemented with 2 mg/ml of BSA.

Furthermore, obtained subcellular fractions were used for protein or RNA isolation. For protein steady-state level analysis, the concentration of the total, cytosolic, and solubilized in IB buffer MB and HS fractions were estimated by Bradford assay or Direct Detect spectrophotometer (Millipore). Protein samples were solubilized in Laemmli buffer with 50 mM DTT, denatured at 65°C for 15 min, and subjected to SDS–PAGE and Western blotting. RNA isolation procedure is described below.

### RNA isolation

RNA was isolated from subcellular fractions using the TRI Reagent solution (AM9738; Thermo Fisher Scientific). 500 $\mu$l of TRI Reagent was mixed with the sample and kept at room temperature for 8 min into each sample, 100 $\mu$l of chloroform was added, mixed and shaken for 15 s, followed by incubation at room temperature for 10 min. The samples were then centrifuged at 12,000$g$ for 15 min at 4°C. Upon transfer of the aqueous phase to a new tube, 250 $\mu$l of isopropanol was added, followed by immediate vortexing for 15 s. The samples were precipitated in −20°C overnight. The next day,

samples were centrifuged at 20,000*g* for 15 min at 4°C. After discarding the supernatant, the pellets were washed with 500 *µ*l of 75% ethanol. The RNA pellets were subsequently dissolved in an appropriate volume of RNase free water and treated with RNase-Free DNase (79254; Qiagen) by incubation at room temperature for 10 min to avoid genomic DNA contamination. Furthermore, the RNA samples were cleaned and concentrated using Agencourt RNA-Clean XP beads (A63987; Beckman Coulter) following the 1.8× reaction volume protocol. The RNA concentration was estimated by Quantus Fluorometer (Promega) or NanoDrop Spectrophotometer (Thermo Fisher Scientific). For library generation, the integrity of RNA was verified by 2200 TapeStation system (Agilent Technologies). The RNA samples were stored in –80°C until further use.

# qRT-PCR

cDNA synthesis was carried out using Verso cDNA synthesis kit (AB-1453/B; Thermo Fisher Scientific) with an input of 250 ng of total RNA. ERCC Spike-In (4456739; Thermo Fisher Scientific) controls (0.5 *µ*l of mix 1 or mix 2 from 1:100 dilution) were added into RNA samples. Random hexamers were used as primers for reverse transcription. Gene-specific primers for qRT-PCR were designed with the help of primer BLAST software (Table S2). qRT-PCR was performed using PowerUp SYBR Green Master Mix (A25742; Thermo Fisher Scientific). The reaction was carried out in the 7900HT Real-Time PCR Instrument (Applied Biosystems) using default standard cycling mode conditions. Relative enrichment of RNA was calculated by normalizing Ct values against endogenous control using the $2^{-\Delta\Delta Ct}$ method (Livak & Schmittgen, 2001). For mitochondria versus MB comparison, *mt-atp8* was used as reference gene. Whereas for RNA-seq validations, an ERCC Spike-In transcript (ERCC-00096) was used as reference.

### Short-read RNA sequencing

As input, 2 *µ*g of RNA from each sample was incorporated with ERCC ExFold RNA Spike-In mixes (4456739; Thermo Fisher Scientific) following the manufacturer's guideline. The libraries were prepared using the KAPA Stranded mRNA-Seq kit (KK8420; Kapa Biosystems, Inc.) as follows. After polyA capture and RNA fragmentation step (6 min at 94°C), the first-strand cDNA was synthesized with random primers. Furthermore, second-strand synthesis was carried out to convert the cDNA:RNA hybrid to double-stranded cDNA along with marking the second strand with dUTP. The samples were then cleaned up using 1.8× Ampure bead-based protocol and immediately proceeded to A-Tailing to add dAMP to 3' end of double-stranded cDNA fragments. After A-Tailing, KAPA single indexed adaptor set A (KK8701; Kapa Biosystems, Inc.) was used for adaptor ligation. Samples were cleaned up twice using Ampure beads (1× bead-based). Next, the adaptor-ligated library DNA was amplified by 11 cycles of PCR. Finally, the library was cleaned up using Ampure beads (1× bead-based). Library fragment size distribution was confirmed by electrophoresis in TapeStation (Agilent Technologies). Library concentration was determined by both Quantus and qPCR (KAPA Library Quantification Kit Illumina Platforms, KR0405). Sequencing was performed on the NextSeq 550 instrument

(Illumina) using the v2 chemistry, resulting in an average of ~100 M reads per library with 2 × 75 bp paired-end setup.

### RNA-seq data analysis

Raw Illumina NextSeq 550 reads were assessed for quality, adapter content, and duplication rates with FastQC (Andrews, 2010). Next, raw Illumina reads were aligned to the reference transcriptome obtained using the masked DanRer11 (GRCz11) genome (plus the sequences of 96 ERCC Spike-In Controls) and Ensembl zebrafish gene annotation (v.101). The transcriptome sequence was prepared using the in house developed script for retrieving spliced transcript sequences. Whereas the short-read mapping and quantification at the gene and isoform level was performed using Salmon (Patro et al, 2017) (v.1.4.0) with default parameters. For the analysis of genome coverage, short-reads were aligned to the masked DanRer11 (GRCz11) genome assembly using STAR (Dobin et al, 2013) (v.2.5.1) compiled for short reads with following non-default parameters: –outFilterMismatchNoverLmax 0.04 –alignIntronMin 20 –alignIntronMax 1000000 –alignMatesGapMax 1000000 –outSAMunmapped Within –runThreadN 6. Illumina RNA-seq mapping statistics are summarized in Fig S1A and B. The genome coverage analysis (Fig S1C and D) was performed using *bamstats* tool developed by Roderic Guigo's group (https://github.com/guigolab/bamstats).

Gene enrichment analysis was performed using DESeq2 (Love et al, 2014) (version 1.30) and by directly comparing MB to HS fractions for WT and *chchd4a*$^{-/-}$ samples, respectively. Only genes with total number of counts >50 across MB and HS samples were included in this analysis. Moreover, genes with a minimum fold change of $\log_2$ <=> ±1 and a maximum Benjamini–Hochberg corrected *P*-value of 0.05 were deemed to be classified as (highly) enriched in each fraction (Table S3). To control the quality of the analysis, we also distinguished moderately enriched genes with a minimum fold change of $\log_2$ <=> ± (0.5–1) and a maximum Benjamini–Hochberg corrected *P*-value of 0.05. The analysis on the transcript level was performed using dominant transcripts—transcripts showing the highest expression levels across all the isoforms for a given gene.

Previously generated RNA-seq data from whole unfractionated *chchd4a*$^{-/-}$ and WT samples were quantified at the gene level using Salmon (Patro et al, 2017) (v.1.4.0) with default parameters. Differentially expressed genes were identified using DESeq2 (Love et al, 2014) (version 1.30). Only genes with a minimum fold change of $\log_2$ <=> 0.9, a maximum Benjamini–Hochberg corrected *P*-value of 0.05, and a minimum combined mean of five reads were deemed to be significantly differentially expressed (Table S4).

### Evolutionary conservation

Human orthologues of zebrafish genes (Table S5) were obtained using BioMart data mining tool from Ensembl. Next, the conservation scores from 100-way PhastCons (Siepel et al, 2005) conservation tracks were extracted using GENCODE (v.30) human exon coordinates corresponding to selected orthologues of zebrafish genes. PhastCons conservation 100 conservation tracks contain scores for the human genome calculated from multiple alignments with other 99 vertebrate species.

## KEGG and GO enrichment analysis

GO and KEGG enrichment analysis for Illumina RNA sequencing data was performed using KEGG.db (v.3.2.3) and GO.db (v.3.4.1) R/Bioconductor packages. The KEGG enrichment analysis was performed using zebrafish data from package org.Dr.eg.db (v 3.4.1). Pathways and genes detected in each fraction were filtered after Benjamini–Hochberg correction for an adjusted $P$-value < 0.05.

## Motif discovery

Homer (Heinz et al, 2010) and STREME (Bailey, 2021) tools were used to extract motifs in the DNA sequence of 3′ and 5′UTR regions of genes enriched in MB and HS fractions in WT and $chchd4a^{-/-}$ samples, respectively. Differential motif discovery using Homer software was performed using *findMotifsGenome.pl* script with default parameters, except –size 200 and –len 10. Whereas analysis using STREME was performed using MEME Suite (v 5.4.1) and shuffled input sequences as a reference.

## RNAscope and immunostaining

Whole-mount fluorescent in-situ hybridization by RNAscope was performed using RNAscope Multiplex Fluorescent v2 Assay kit (Advanced Cell Diagnostics) according to the manufacturer's guideline with the following changes. 5 dfp larvae in the *Tg(Xla.Eef1a1:mlsEGFP)cms1* were fixed with 4% paraformaldehyde for 18 h at RT. The larvae were depigmented using a solution of 3% $H_2O_2$ and 1% KOH in $dH_2O$ for 10 min at RT. Furthermore, the samples were washed in 1×PBS + 0.1% Tween 20 (PBST), serially dehydrated and stored in 100% methanol at –20°C until use. Next, the samples were treated with 0.2 M HCl in 100% methanol for 10 min at RT and rehydrated in decreasing concentrations of methanol in 1×PBST. Upon final wash with PBST +1% BSA, the samples were transferred to a preheated target retrieval solution and heated in a thermoblock for 15 min at 98°C. Immediately, the samples were immersed in PBST +1% BSA for 1 min, washed with 100% methanol and then again suspended in PBST +1% BSA. Further protease digestion was performed by treating the samples with Protease Plus reagent for 15 min in a preheated water bath at 40°C. This step was followed by the probe hybridization for 2 h at 40°C. Hybridization with AMP1, AMP2, and AMP3 solutions and development of HRP signal were done as per the protocol. Samples were treated with a single probe at a time and in combination with TSA Plus Cyanine5 (PerkinElmer) at a dilution of 1:1,500. We used *myod1* probe and custom-made *mt-nd5*, *rcn2*, *slc16a1a*, and *comtd1* probes (Advanced Cell Diagnostics). Owing to the quenching of endogenous GFP signal, the RNAscope protocol was followed by immunostaining with anti-GFP antibody (GTX113617). Briefly, the samples were washed with PBST and blocked with a buffer containing 10% goat serum, 1% DMSO, 0.5% Triton X-100 in PBST for 1 h at RT with gentle shaking. The primary antibody was diluted 1:200 in the blocking buffer and the samples were incubated overnight at 4°C. Next, the samples were washed thrice with PBST for 10 min at RT and incubated with Alexa Fluor 488–coupled donkey anti-rabbit secondary antibody (Thermo Fisher Scientific) diluted 1:500 in blocking buffer, for 1 h at RT. After washing thrice with PBST for 10 min at RT, the DNA was counterstained with DAPI (4′,6-diamidino-2-phenylindole; Thermo Fisher Scientific) diluted 1:10,000 with PBST, for 40 min at RT. The larvae were mounted on glass slides with specially created cavities using Prolong Gold Antifade mountant (Thermo Fisher Scientific). Images were acquired using LSM800 confocal laser scanning microscope (Zeiss). Region of interest defined by a dimension of 700 × 250 $\mu$m surrounding DAPI staining was chosen from three different biological replicates and Mander's M1 colocalization coefficient defining the extent of mRNA fraction colocalized with mitochondria was calculated with the Manders Coefficients ImageJ plugin.

# Data Availability

All computer code is available from the authors upon request. Sequence data have been deposited in the Gene Expression Omnibus repository under accession number GSE167587.

# Supplementary Information

# Acknowledgements

We thank M Macias and T Wegierski (IIMCB) for assistance with microscopy and RNA localization analysis. We show appreciation to R Guigo (CRG), U Topf, and M Turek (from the Chacinska group at the time of this study) for fruitful discussions and M Diekhans (University of California, Santa Cruz) and R Suratekar (IIMCB) for the help with data analysis. We thank J Kuznicki (IIMCB) for the laboratory and scientific support. We also acknowledge the IIMCB Zebrafish Core Facility for service and fish material. We are grateful to members of the Chacinska, Uszczynska-Ratajczak, and Winata labs for useful insights. This work was supported by the POLONEZ Fellowship of National Science Centre, Poland, 2016/23/P/NZ3/03730. This project has received funding from the European Union's Horizon 2020 research and innovation programme under the Marie Skłodowska-Curie grant agreement No 665778 (B Uszczynska-Ratajczak). This work was also supported by EU/FP7: Research Potential FISHMED, grant number 316125. The work was also funded by "Regenerative Mechanisms for Health" project MAB/2017/2 carried out within the International Research Agendas program of the Foundation for Polish Science cofinanced by the European Union under the European Regional Development Fund.

## Author Contributions

B Uszczynska-Ratajczak: conceptualization, resources, data curation, software, formal analysis, supervision, funding acquisition, validation, investigation, visualization, methodology, project administration, and writing—original draft, review, and editing.
S Sugunan: resources, validation, investigation, methodology, and writing—original draft, review, and editing.
M Kwiatkowska: resources, data curation, investigation, methodology, and writing—review and editing.
M Migdal: formal analysis, investigation, visualization, and writing—review and editing.

S Carbonell-Sala: data curation, investigation, and writing—review and editing.

A Sokol: investigation, methodology, and writing—review and editing.

CL Winata: conceptualization, supervision, and writing—review and editing.

A Chacinska: conceptualization, supervision, funding acquisition, project administration, and writing—original draft, review, and editing.

## Conflict of Interest Statement

The authors declare that they have no conflict of interest.

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
