## [Reviewer comments · Life Science Alliance]

Life Science Alliance

Profiling subcellular localization of nuclear-encoded mitochondrial gene products in zebrafish

Barbara Uszczynska Ratajczak, Sreedevi Sugunan, Monika Kwiatkowska, Maciej Migdal, Silvia Carbonell-Sala, Anna Sokol, Cecilia Winata, and Agnieszka Chacinska

DOI: <https://doi.org/10.26508/lsa.202201514>

Corresponding author(s): Agnieszka Chacinska, The International Institute of Molecular Mechanisms and Machines Polish Academy of Sciences and Barbara Uszczynska Ratajczak, Institute of Bioorganic Chemistry Polish Academy of Sciences

Review Timeline:

Submission Date:	2022-05-04
Editorial Decision:	2022-06-12
Revision Received:	2022-09-03
Editorial Decision:	2022-09-22
Revision Received:	2022-09-30
Accepted:	2022-10-04

Scientific Editor: Novella Guidi

Transaction Report:

June 12, 2022

Re: Life Science Alliance manuscript #LSA-2022-01514-T

Agnieszka Chacinska

The International Institute of Molecular Mechanisms and Machines, Polish Academy of Sciences, Warsaw, Poland

Dear Dr. Chacinska,

Thank you for submitting your manuscript entitled "Profiling subcellular localization of nuclear-encoded mitochondrial gene products in zebrafish" to Life Science Alliance. The manuscript was assessed by expert reviewers, whose comments are appended to this letter. We invite you to submit a revised manuscript addressing the Reviewer comments.

Thank you for this interesting contribution to Life Science Alliance. We are looking forward to receiving your revised manuscript.

Sincerely,

B. MANUSCRIPT ORGANIZATION AND FORMATTING:

Reviewer #1 (Comments to the Authors (Required)):

Mitochondria import hundreds of different proteins from the cytosol. The timing of protein synthesis and protein translocation is not well understood. Previous studies proposed that the large majority of precursors is translocated after completion of their synthesis (i. e. post-translationally). However, some reports (including studies by the authors of this manuscript) showed evidence for the co-translational translocation of specific proteins, in particular of hydrophobic inner membrane proteins. This present study presents a very comprehensive and broad analysis of transcripts which co-purify with mitochondrial (and microsomal) membranes, using zebrafish as a model. Since most previous work was done in yeast, this is clearly a very interesting model system to address such a basic question. The authors describe here a novel fractionation protocol to isolate mitochondrion-free high speed pellet sample, they managed to distinguish mitochondria-bound from ER-bound transcripts. Next, they analyzed which proteins are made on the surface of mitochondria, identifying elements on the mRNA level (sequence motifs) and features of the synthesized nascent polypeptides (length and hydrophobicity). Finally, they studied a Mia40 mutant in order to test whether defects in the import machinery influence ribosome-binding.

This is a very interesting study which shows that in fish, similar to the situation in yeast, mitochondrial precursors are imported predominantly post-translationally. The data are very clear and compelling. Before this study is published, the authors might take care for a couple of points:

1. The graphical analysis of the data sets could be improved. The authors sort their candidate genes into categories and then discuss these categories in depth. Thereby, only very general features will be described and it is impossible for readers to see the behavior of individual data points. They might consider to define a score by which proteins could be individually ranked. For example, an enrichment score, indicating how much of given protein was found in the mitochondrial fraction. Thereby, they could define a leading group of proteins, for which co-translational targeting is indeed likely.
2. Such enrichment scores were calculated before in a study of the Weissman lab (Williams et al., cited as ref. 20). In that study, proteins such as ABC transporters (Atm1, Mdl1, Mdl2) or other inner membrane proteins (Yme1, Yme2, Yta10, Yta12) were suggested to be co-translationally imported in yeast. The authors should compare their 'top group' to the results of the Weissman paper to see whether similar proteins behave similarly in yeast and fish. Presence of hydrophobic transmembrane domains is a very poor indicator also on the Weissman list as most inner membrane proteins were found as not being made by mito-bound ribosomes.
3. The observation that longer proteins are more likely to bind co-translationally to mitochondria is interesting, though not unexpected. A longer synthesis time just increases the time for a presequence to bind the TOM complex. Again, a correlation score would be interesting here. The authors might just plot length versus enrichment on mitochondria. This should distinguish a gradual increase due to kinetics to their hypothesis that different classes of precursors exist which use different import modes.
4. The study starts with a nice overview in Figure 1A. However, at the end of the figures, there is no graphical summary of the observations. I propose to make a graph to illustrate the major take-home message of this study. This will increase the visibility of the study and the probability to be cited in other studies on co-vs-post-targeting in the future.

Minor points

5. Page 16: 'Proteins encoded by these three genes were also showed to be localizing to mitochondria by numerous studies'. Replace showed by shown.
6. Page 18: 'transcripts (median = 538 nt vs. median = 496 nt) Surprisingly, the length of CDSs in other': Please insert a full stop.
7. Page 20: 'Many studies reported the presence of mRNAs on the mitochondrial surface in yeast and human cell lines, but so far no such analysis has been performed in vivo in higher eukaryotes'. This sentence should be rephrased as humans are higher eukaryotes. In any case, studying the binding of ribosomes to mitochondria in the zebrafish model is per se interesting.
8. Page 22: 'there is evidence for localization of SLC16A1 into mitochondria.' Replace into by in.

Reviewer #2 (Comments to the Authors (Required)):

In his study the authors present the development of a biochemical fractionation method to obtain membrane-bound fraction containing intact mitochondria and ER with ribosomes on their surface. The aim of the study was to investigate the presence of mRNAs for mitochondrial proteins on the surface of the organelle in vivo, as this was done previously only in *S cerevisiae* and in mammalian cells in culture. Further, the authors focused on one of the main protein import pathways (the IMS pathway requiring the

protein Mia40) and expanded their previous analysis of changes induced in zebrafish model by mutating the key component of this pathway. The paper is mainly of interest as a methodological advance and could be of interest as an additional genomic analysis approach for the process of mitochondria biogenesis in a whole animal. I have some points I think the authors can address to clarify certain aspects of their work:

1. There are already reported studies on semipermeabilized cells (even yeast cells) as a method to analyze presence of mRNAs (and the protein import process), which overcome the caveats of using isolated organelles. So to some extent these approaches allow analysis of ER-mito contact sites and preferential association of mRNAs so the authors need to comment on this and more clearly present the advantages of their approach. The main question that arises in such methodological advances is how the zebrafish model promotes our understanding of basic mechanisms.
2. top of p. 11 and Fig 1C: please explain the role of BSA, as it is not clear how it inhibits proteolysis
3. Fig 3A: presence of ABC transporters and SLC family members (solute carriers) in the MB and HS fraction. It is not clear why these would be enriched (particularly the ABC transporters), and other TMD containing proteins (like 7 TMD proteins) are not. Mitochondria are not known to contain increased levels of ABC transporters so there must be some link to function which the authors need to comment on.
4. Fig5A: the authors show that a number of pathways are affected in the Mia40 mutation strain, but then focus on discussing only a number of pathways (amino acid activation and protein refolding). There are many other pathways (several metabolic pathways) that are affected and although the refolding and amino acid activation pathways are to some extent expected to be affected the others that are not discussed are more unexpected. It would be worthwhile to expand this discussion.

Reviewer #1 (Comments to the Authors (Required)):

Mitochondria import hundreds of different proteins from the cytosol. The timing of protein synthesis and protein translocation is not well understood. Previous studies proposed that the large majority of precursors is translocated after completion of their synthesis (i. e. post-translationally). However, some reports (including studies by the authors of this manuscript) showed evidence for the co-translational translocation of specific proteins, in particular of hydrophobic inner membrane proteins. This present study presents a very comprehensive and broad analysis of transcripts which co-purify with mitochondrial (and microsomal) membranes, using zebrafish as a model. Since most previous work was done in yeast, this is clearly a very interesting model system to address such a basic question. The authors describe here a novel fractionation protocol to isolate mitochondrial membranes. These membrane fractions also contain microsomes. However, by comparison to a rather mitochondrion-free high speed pellet sample, they managed to distinguish mitochondria-bound from ER-bound transcripts. Next, they analyzed which proteins are made on the surface of mitochondria, identifying elements on the mRNA level (sequence motifs) and features of the synthesized nascent polypeptides (length and hydrophobicity). Finally, they studied a Mia40 mutant in order to test whether defects in the import machinery influence ribosome-binding.

This is a very interesting study which shows that in fish, similar to the situation in yeast, mitochondrial precursors are imported predominantly post-translationally. The data are very clear and compelling. Before this study is published, the authors might take care for a couple of points:

We thank the Reviewer for positive comments and insightful feedback.

1. The graphical analysis of the data sets could be improved. The authors sort their candidate genes into categories and then discuss these categories in depth. Thereby, only very general features will be described and it is impossible for readers to see the behavior of individual data points. They might consider to define a score by which proteins could be individually ranked. For example, an enrichment score, indicating how much of given protein was found in the mitochondrial fraction. Thereby, they could define a leading group of proteins, for which co-translational targeting is indeed likely.

We fully acknowledge the importance of this point. Although the main goal of this study at the level of data analysis was to create an enrichment score allowing to rank the zebrafish genes by their subcellular localization, we fully agree that this intention was not clearly explained in the text. To improve the clarity of the text, we added a sentence highlighting this fact:

“To rank transcripts from the most membrane associated to those most likely translated by the free cytosolic polysomes, we directly compared two fractions with each other. The HS fraction was a reference in these comparisons, so all genes that are enriched in this analysis are likely to be membrane bound (MB-enriched) and those that are depleted are more likely to be translated by free cytosolic polysomes (HS-enriched).” (Page 12).

We also aimed to define a leading group of genes/proteins for which co-translational targeting is either very likely or unlikely. For this reason we performed the Kyoto Encyclopaedia of Genes and Genomes (KEGG) enrichment analysis. However, as described previously this analysis revealed only enrichment of ABC transporters and genes involved in oxidative

phosphorylation in the membrane-bound (MB) and high-speed fraction, respectively. Although zebrafish is the third best annotated genome, many zebrafish genes lack functional annotation. To overcome this problem and extend the analysis of MB- and HS-enriched gene classes, we also performed Gene Ontology enrichment analysis using highly and moderately enriched genes identified for each of studied fractions using two different GO aspects: “biological process” and “cellular component” (Figure 2B, Figure S2C-F and Figure S8D-E).

2. Such enrichment scores were calculated before in a study of the Weissman lab (Williams et al., cited as ref. 20). In that study, proteins such as ABC transporters (Atm1, Mdl1, Mdl2) or other inner membrane proteins (Yme1, Yme2, Yta10, Yta12) were suggested to be co-translationally imported in yeast. The authors should compare their 'top group' to the results of the Weissman paper to see whether similar proteins behave similarly in yeast and fish. Presence of hydrophobic transmembrane domains is a very poor indicator also on the Weissman list as most inner membrane proteins were found as not being made by mito-bound ribosomes.

Once again this is a very insightful comment. As the Weissman group's study targeting the plasticity of mitochondrial proteins solely concentrated on mitochondria, excluding ER from this analysis, we selected yeast genes with the at least two fold mitochondrial enrichment and tested them against the enrichment values for their zebrafish orthologues. The overall correlation was rather poor, as this comparison revealed the presence of two subpopulations: MB and HS enriched zebrafish genes. However, indeed, we observed zebrafish orthologues (*abcb11a*, *abcb11b* and *tap1*) of MDL1 and MDL2 to be highly enriched in the MB fraction. Moreover, we also noticed high MB-enrichment of genes involved in ion transport, e.g. zebrafish solute carriers (*slc30a1a* and *slc30a8*) that are orthologous to MMT and MMT2 in yeast. Interestingly, the overlap between MB-enriched zebrafish genes and yeast genes translated on the surface of mitochondria was higher in cells without CHX compared to CHX-treated ones (Figure S4). This effect holds also for comparison using *chchd4a*^{-/-} fractions (Figure S9).

We previously showed that the zebrafish orthologue (*abcb7*) of Atm1 was moderately enriched in the MB-fraction ($\log_2\text{FC MB/HS} = 0.5$) (Figure 2E). In general, we were not able to provide reliable enrichment scores for other inner membrane yeast proteins (Yme1, Yme2, Yta10, Yta12). Although we found zebrafish orthologues for Yme1 gene (ENSDARG00000075192, *yme11a* and ENSDARG00000104401, *yme11b*) these genes were not showing major changes in the localization ($\log_2\text{FC} = -0.39$ and $\log_2\text{FC} = -0.069$, respectively). Moreover, the obtained enrichment results were not statistically significant. For both Yta10 and Yta12 genes, we detected the same zebrafish orthologues (ENSDARG0000007965, *afg311* and ENSDARG00000062272, *afg312*). There are no major changes in the localization for *afg312* ($\log_2\text{FC} = -0.062$), but again the enrichment is not statistically sufficient. Whereas, the *afg311* is more HS enriched ($\log_2\text{FC} = -0.75$, FDR=1.3%), but the result is below the cutoff point for highly enriched genes ($\log_2\text{FC} = < -1$). Finally, we were not able to find zebrafish orthologues for the Yme2 gene.

3. The observation that longer proteins are more likely to bind co-translationally to mitochondria is interesting, though not unexpected. A longer synthesis time just increases the time for a presequence to bind the TOM complex. Again, a correlation score would be interesting here. The authors might just plot length versus enrichment on mitochondria. This should distinguish a gradual increase due to kinetics to their hypothesis that different classes of precursors exist which use different import modes.

Thank you for this very interesting and important comment. We followed Reviewer's suggestion and plotted the spliced CDS length versus the enrichment in the MB and HS fraction. Although we did not observe big difference when looking at all genes for wild type fractions (Figure S11A), there is a large shift in the CDS length for *chchd4a*^{-/-} samples, as MB-enriched transcripts have on average much longer CDSs than the ones enriched in HS fraction (Figure S11B). This effect also holds when looking only at genes encoding mitochondrial proteins (Figure S11C-D). However, the CDS length increases gradually with the MB-enrichment, in particular when looking at all enriched genes (Figure S11B) and it is not possible to set any specific threshold that could help to distinguish the MB-enriched transcripts from HS-enriched ones. Moreover, there is a link between the sequence conservation and its length, as highly conserved proteins have on average longer sequences. We confirm this finding by looking at zebrafish and human protein coding genes (Figure S11E-F). Thus, there are many factors that could drive the co-translational import of large proteins. However, it could be that the link between the protein size, protein conservation and prolonged time of the synthesis is an important aspect determining the choice of the protein import route.

Appropriate text explaining these findings has been added to the Results section (Page 23) and Discussion (Page 28).

4. The study starts with a nice overview in Figure 1A. However, at the end of the figures, there is no graphical summary of the observations. I propose to make a graph to illustrate the major take-home message of this study. This will increase the visibility of the study and the probability to be cited in other studies on co-vs-post-targeting in the future.

We very much appreciate this excellent idea. The figure summarizing the main findings of this study is now available as Figure 7.

Minor points

5. Page 16: 'Proteins encoded by these three genes were also showed to be localizing to mitochondria by numerous studies'. Replace showed by shown.

Fixed.

6. Page 18: 'transcripts (median = 538 nt vs. median = 496 nt) Surprisingly, the length of CDSs in other': Please insert a full stop.

Fixed.

7. Page 20: 'Many studies reported the presence of mRNAs on the mitochondrial surface in yeast and human cell lines, but so far no such analysis has been performed *in vivo* in higher eukaryotes'. This sentence should be rephrased as humans are higher eukaryotes. In any case, studying the binding of ribosomes to mitochondria in the zebrafish model is per se interesting.

Changed to:

"Many studies reported the presence of mRNAs on the mitochondrial surface in yeast and *in vitro* human cell lines, but so far no such analysis has been performed *in vivo* in higher eukaryotes".

8. Page 22: 'there is evidence for localization of SLC16A1 into mitochondria.' Replace into by in.

Done.

Reviewer #2 (Comments to the Authors (Required)):

In his study the authors present the development of a biochemical fractionation method to obtain membrane-bound fraction containing intact mitochondria and ER with ribosomes on their surface. The aim of the study was to investigate the presence of mRNAs for mitochondrial proteins on the surface of the organelle in vivo, as this was done previously only in *S cerevisiae* and in mammalian cells in culture. Further, thus authors focused on one the main protein import pathways (the IMS pathway requiring the protein Mia40) and expanded their previous analysis of changes induced in zebrafish model by mutating the key component of this pathway. The paper is mainly of interest as a methodological advance and could be of interest as an additional genomic analysis approach for the process of mitochondria biogenesis in a whole animal. I have some points I think the authors can address to clarify certain aspects of their work:

We wish to thank the Reviewer for these positive comments and helpful suggestions for improvement of the manuscript.

1. There are already reported studied on semipermeabilized cells (even yeast cells) as a method to analyze presence of mRNAs (and the protein import process), which overcome the caveats of using isolated organelles. So to some extent these approaches allow analysis of ER-mito contact sites and preferential association of mRNAs so the authors need to comment on this and more clearly present the advantages of their approach. The main question that arises in such methodological advances is how the zebrafish model promotes our understanding of basic mechanisms.

We appreciate this suggestion. Appropriate text has been added to the discussion (Page 25):

*“Recent technological advancements in the RNA sequencing field also triggered a substantial progress in studying RNA at subcellular resolution. Currently, there are two main methods for producing quantitative subcellular RNA maps: (1) “CeFra-seq” developed within the ENCODE project, where the cells are fractionated to extract RNA from specific compartments and (2) the APEX-seq method, using transgenic labelled proteins of known localization to cross-link nearby RNAs. Although both methods are very versatile and can capture the landscape of all RNA types (coding and noncoding) for any subcellular region in the cell, the application of APEX-seq in vivo is more challenging. This method employs APEX2 peroxidase that is genetically targeted to the cellular region of interest to tag endogenous RNAs using proximity biotinylation. Thus, requires genetic manipulations that allow to incorporate these elements into the genome of studied organism. Although similar genetic modifications have been previously described for zebrafish, in general production of transgenic lines for vertebrate species is not only much more challenging and time consuming, but also often requires specialized equipment. New genetic modifications are particularly problematic for mutant lines, especially if the mutation inhibits the development or turns out to be lethal in its early stage, as in case of the *chchd4a*^{-/-} mutation in zebrafish. Another possibility is a proximity specific ribosome profiling that enables investigation of localized translation genome-wide at subcellular resolution. It has been mainly employed to profile in vivo actively translated mRNAs on the endoplasmic reticulum membrane and on the surface of mitochondria. However,*

this method employs a spatially restricted biotin ligase (BirA) to label ribosomes with a biotin acceptor peptide (AviTag) in live cells. Therefore, limitations of adopting this strategy for zebrafish are similar as for the APEX-seq approach. Finally, other methods for direct analysis of RNA localization at high-throughput exist, including highly multiplex RNA profiling. However, these methods require designing thousands of barcoded oligonucleotide probes to target RNA of interest. Moreover, they can only measure a limited number of RNA molecule types in a single experiment, which often does not exceed 1,000 RNA types. One of the major drawbacks of all fluorescence in situ hybridization based approaches is the need for cell fixation and permeabilization that can lead to the displacement of cellular components. Considering the above, biochemical fractionation is still one of the most accessible approaches for studying the subcellular localization of nuclear-encoded mitochondrial gene products in zebrafish.” (All of these statements are properly referenced in the main text.)

Although we fully agree that the current technological advancements resulted in the development of new and more specific technologies for studying subcellular RNA localization at high-throughput, it is still difficult to directly adopt them for studying RNA localization *in vivo* using vertebrate models. Especially, if one aims to study RNA localization under mitochondrial stressed conditions triggered by the mutation. We developed two transgenic lines with (1) biotin ligase (BirA) fused with TOM20, an outer mitochondrial membrane protein and (2) Avi-tagged ribosomal lines. However, crossing them together was very challenging, as well as generating them for *chchd4a*^{-/-} zebrafish mutants.

2. top of p. 11 and Fig 1C: please explain the role of BSA , as it is not clear how it inhibits proteolysis.

Thank you for pointing this out. Indeed this is an editorial mistake that distorted the meaning of the entire statement. The proper explanation is available on Page 11:

“BSA helps to preserve function and integrity of mitochondria and is expected to enable reliable profiling of MB-associated RNAs. It actually neutralizes the negative action of fatty acids activated by phospholipases during cell disruption, which inhibits proper function of mitochondria by uncoupling oxidative phosphorylation. At high concentration, BSA may protect mitochondrial proteins from degradation, serving as an alternative substrate for proteases released upon cell breakage.” (All of these statements are properly referenced in the main text.)

3. Fig 3A: presence of ABC transporters and SLC family members (solute carriers) in the MB and HS fraction. It is not clear why these would be enriched (particular the ABC transporters), and other TMD containing proteins (like 7 TMD proteins) are not. Mitochondria are not known to contain increased levels of ABC transporters so there must be some link to function which the authors need to comment on.

This is a very insightful comment. Previous studies, including Weissman’s proximity specific ribosome profiling at the surface of mitochondria revealed the mitochondrial enrichment of ABC transporters such as *ATM1*, *MDL1* and *MDL2*. Moreover, this data supported enrichment of ion transporters such as *MMT1*, *MMT2* and *OAC1*. Interestingly, a recently published APEX-seq study revealed the *SLC39A10* gene to be located in the close proximity to the mitochondrial outer membrane. Similarly, our data supported the enrichment of genes encoding ABC transporters and solute carriers. As we observed rather moderate MB-enrichment of mitochondrial ABC transporter genes, we can openly admit that detected membrane

association of ABC transporters could be also ER driven. To seek for the possible link to a function that could specifically drive the MB enrichment of ABC transporters and SLC family members, we investigated the enrichment for proteins containing other TMDs, in particular 7TMD proteins. In this analysis we concentrated on the G protein-coupled receptor (GPCR) superfamily (Figure S5C). We were able to detect 10 GPCRs in our data and interestingly 6 and 4 of them were highly and moderately enriched, respectively in the MB fraction. Furthermore, we tested the enrichment for Cytochrome P450 proteins (CYPs) as some of them (*cyp20a1*) were shown to localize to mitochondria. We were able to detect 42 CYP genes and all of them, except *cyp17a1* were highly MB enriched (Figure S5A). In general, CYP localization in mitochondria is regulated in one of two ways: (1) direct targeting of inherent CYPs with canonical mitochondrial signals in their protein sequence after synthesis in the cytosol or (2) mitochondrial localization of microsomal CYPs after processing of the NH(2)-terminal region. Therefore, it seems that all these genes can use distinct mechanisms to localize to mitochondria (or ER). Although there can exist a functional link for ABC transporters and SLC genes driving their mitochondrial localization, this will require further investigation, as this study lacks the power to properly answer this question. Therefore, we stick with the general mechanism that coupling import to translation can protect transmembrane proteins from being randomly incorporated to other membranes. Finally, absence of some of the gene classes in both KEGG and GO enrichment analysis confirms that poor functional annotation of zebrafish genes hampers analysis at the global level. Moreover, despite many attempts to fix this problem, GO remains messy with not fully controlled assignment of genes to parent and child GO terms.

4. Fig5A: the authors show that a number of pathways are affected in the Mia40 mutation strain, but then focus on discussing only a number of pathways (amino acid activation and protein refolding). There are many other pathways (several metabolic pathways) that are affected and although the refolding and amino acid activation pathways are to some extent expected to be affected the others that are not discussed are more unexpected. It would be worthwhile to expand this discussion.

We fully acknowledge the importance of this point. Following Reviewer's suggestion the discussion has been expanded and the possible link between *chchd4a*^{-/-} mutation in zebrafish and the reduced proteolysis and upregulated cholesterol biosynthesis have been explained (Page 20):

*“As disruption of mitochondrial functions leads to widespread consequences, we also observed expression changes for genes involved in several metabolic pathways such as proteolysis and cholesterol biosynthetic processes. As previously described *chchd4a*^{-/-} mutation leads to pancreatic insufficiency due to reduced expression of various digestive enzymes and their precursors, which affects the overall development of pancreas and as a result to inability of *chchd4a*^{-/-} mutants to digest and absorb proteins. Whereas, the current analysis revealed an additional link between *chchd4a*^{-/-} mediated mitochondrial insufficiency and the upregulation of cholesterol biosynthesis. Mitochondrial dysfunction has been shown to be associated with increased accumulation of cholesterol in the cell, as mitochondria play a crucial role in cholesterol metabolism. Moreover, sterols play a role in mitochondrial retrograde signalling, which determines the way mitochondria contact with the rest of the cell. As a consequence cholesterol synthesis is strictly regulated by the endoplasmic reticulum, in particular its membrane proteins possessing the ability to sense cholesterol levels. Impairing mitochondrial*

*functions disrupts retrograde signalling, which in turns affects the distribution of free cholesterol in the cell and finally the ability of ER proteins to properly sense sterol levels. Elevated cholesterol levels have been shown to downregulate mevalonate pathway genes, reducing the biosynthesis of endogenous cholesterol and as a result disrupting the final metabolic outcome of this pathway. Interestingly, one of the upregulated genes upon *chchd4a* /mutation is *hmgcs1* encoding a key enzyme in the mevalonate pathway. Recently, *HMGCS1* its human orthologue, has been shown to activate unfolded protein response (UPR) components (*ATF4*, *XBP1* and *ATF6*) and protect mitochondria and ER from the damage under proteostatic stress. Increased cholesterol biosynthesis can also act as a compensatory mechanism reducing redox stress by consuming elevated NAD(P)H levels in the cell.” (All of these statements are properly referenced in the main text.)*

September 22, 2022

RE: Life Science Alliance Manuscript #LSA-2022-01514-TR

Prof. Agnieszka Chacinska
University of Warsaw
Centre of New Technologies
Banacha 2c
Warsaw 02-097
Poland

Dear Dr. Chacinska,

Thank you for submitting your revised manuscript entitled "Profiling subcellular localization of nuclear-encoded mitochondrial gene products in zebrafish". We would be happy to publish your paper in Life Science Alliance pending final revisions necessary to meet our formatting guidelines.

- please add ORCID ID for both corresponding authors-you should have received instructions on how to do so
- please add the Twitter handle of your host institute/organization as well as your own or/and one of the authors in our system
- please consult our manuscript preparation guidelines <https://www.life-science-alliance.org/manuscript-prep> and make sure your manuscript sections are in the correct order
- please use the [10 author names, et al.] format in your references (i.e. limit the author names to the first 10)
- please add the panels E and F to your Figure 4 legend; please add the panel C to your Figure S6 legend
- please add a separate Data Availability section at the end of Materials and Methods for your RNA seq data including your accession code

A. FINAL FILES:

B. MANUSCRIPT ORGANIZATION AND FORMATTING:

Sincerely,

Reviewer #1 (Comments to the Authors (Required)):

The authors satisfactorily addressed all points raised on the initial submission. I fully support publication of this interesting study in its present form.

Reviewer #2 (Comments to the Authors (Required)):

The authors have done several modifications to the ms in response to my comments and those of other reviewers. The revised version addressed all the points I raised and I am happy to recommend its acceptance.

October 4, 2022

RE: Life Science Alliance Manuscript #LSA-2022-01514-TRR

Prof. Agnieszka Chacinska
The International Institute of Molecular Mechanisms and Machines Polish Academy of Sciences
B. Smetany 2
Warsaw 00-783
Poland

Dear Dr. Chacinska,

Thank you for submitting your Research Article entitled "Profiling subcellular localization of nuclear-encoded mitochondrial gene products in zebrafish". It is a pleasure to let you know that your manuscript is now accepted for publication in Life Science Alliance. Congratulations on this interesting work.

DISTRIBUTION OF MATERIALS:

Again, congratulations on a very nice paper. I hope you found the review process to be constructive and are pleased with how the manuscript was handled editorially. We look forward to future exciting submissions from your lab.

Sincerely,
